# ZO-1 Guides Tight Junction Assembly and Epithelial Morphogenesis via Cytoskeletal Tension-Dependent and -Independent Functions

**DOI:** 10.3390/cells11233775

**Published:** 2022-11-25

**Authors:** Alexis J. Haas, Ceniz Zihni, Susanne M. Krug, Riccardo Maraspini, Tetsuhisa Otani, Mikio Furuse, Alf Honigmann, Maria S. Balda, Karl Matter

**Affiliations:** 1UCL Institute of Ophthalmology, University College London, London EC1V 9EL, UK; 2Clinical Physiology/Nutritional Medicine, Charité-Universitätsmedizin Berlin, Campus Benjamin Franklin, 12203 Berlin, Germany; 3Max Planck Institute of Molecular Cell Biology and Genetics, 01309 Dresden, Germany; 4Division of Cell Structure, National Institute for Physiological Sciences, Okazaki 444-8787, Aichi, Japan; 5Biotechnologisches Zentrum (BIOTEC), Center for Molecular and Cellular Bioengineering (CMCB), Technische Universität Dresden, 01307 Dresden, Germany

**Keywords:** tight junction, myosin, F-actin, stiffness, ECM, tension, freeze fracture, occludin, claudin, transepithelial electrical resistance

## Abstract

Formation and maintenance of tissue barriers require the coordination of cell mechanics and cell–cell junction assembly. Here, we combined methods to modulate ECM stiffness and to measure mechanical forces on adhesion complexes to investigate how tight junctions regulate cell mechanics and epithelial morphogenesis. We found that depletion of the tight junction adaptor ZO-1 disrupted junction assembly and morphogenesis in an ECM stiffness-dependent manner and led to a stiffness-dependant reorganisation of active myosin. Both junction formation and morphogenesis were rescued by inhibition of actomyosin contractility. ZO-1 depletion also impacted mechanical tension at cell-matrix and E-cadherin-based cell–cell adhesions. The effect on E-cadherin also depended on ECM stiffness and correlated with effects of ECM stiffness on actin cytoskeleton organisation. However, ZO-1 knockout also revealed tension-independent functions of ZO-1. ZO-1-deficient cells could assemble functional barriers at low tension, but their tight junctions remained corrupted with strongly reduced and discontinuous recruitment of junctional components. Our results thus reveal that reciprocal regulation between ZO-1 and cell mechanics controls tight junction assembly and epithelial morphogenesis, and that, in a second, tension-independent step, ZO-1 is required to assemble morphologically and structurally fully assembled and functionally normal tight junctions.

## 1. Introduction

Preservation of epithelial and endothelial barrier function during tissue morphogenesis and homeostasis requires coordination of cell–cell interaction, ECM adhesion, and cellular mechanical properties. Such adaptive processes rely on cells sensing their physical environment through mechanotransduction pathways activated at adhesion sites that guide cell-wide changes through still not well understood mechanisms that regulate cell morphogenesis and behaviour [1,2,3].

Adhesion complexes are composed of transmembrane adhesion proteins interacting with cytosolic components that serve as linkers to the cytoskeleton and have different types of regulatory functions. The actomyosin cytoskeleton plays an important role in cells’ responses to mechanical stress and in the regulation of adhesion complexes at the cell–cell as well as cell-ECM interface [2]. For example, increasing ECM stiffness stimulates increased cell spreading through mechanotransduction at cell-ECM adhesion sites called focal adhesions [3,4]. Similarly, interplay between different adhesion sites is thought to guide cell and tissue morphogenesis [5,6]. Vertebrate epithelial and endothelial cells adhere to each other via intercellular junctions with two of them, tight (TJ) and adherens (AJ) junctions, forming the apical junctional complex [7,8]. TJ form paracellular semipermeable barriers that discriminate solutes on the basis of size and charge, and are essential for tissues to form functional barriers [8]. Maintenance of the junctional barrier requires TJ to adapt to mechanical stress and cells to respond to forces acting on TJ [9]. However, such processes are still poorly understood.

A central role in the regulation of mechanosensation and force transmission at TJ is played by ZO-1, a multidomain cytosolic protein of the MAGUK family that links the junctional membrane to the cytoskeleton [9,10,11,12]. ZO-1 and the related MAGUK ZO-2 are thought to exist in an open and a closed conformation that differ in their ligand binding properties, a switch thought to be regulated by actomyosin activity [13,14]. Mechanical load on ZO-1 is regulated by the junctional adhesion molecule JAM-A as well as ECM stiffness [9]. JAM-A depletion stimulates junctional actomyosin activation by p114RhoGEF, which is important for junctional integrity. Hence, ZO-1 may be important for junctional integrity when cell–cell contacts carry a high mechanical load. A mechanical load-dependent role of ZO-1 in junctional integrity might also explain the differential impact of ZO-1 depletion on junctional integrity and morphogenetic responses in different experimental models [11,12,15,16,17,18,19,20,21]. ZO-1 knockout mice develop disorganised notochords and neural tubes around an embryonic stage at which these tissues represent the stiffest areas of the developing embryo [18,22]. However, such a stiffness-dependent role of ZO-1 in TJ formation and tissue morphogenesis has not yet been demonstrated.

Here, we combined approaches to tune ECM stiffness to manipulate cell spreading and cytoskeletal tension with methods to measure forces on adhesion complexes to investigate the role of ZO-1 in regulating cell mechanics and epithelial morphogenesis, and to determine how changes in cytoskeletal tension feedback to tight junctions. We found that ZO-1 regulates forces acting on AJ and ECM adhesion but that the impact on AJ was dependent on ECM stiffness, possibly due to differential organisation of the actin cytoskeleton. The requirements for ZO-1 for junction formation and 2D morphogenesis were also tension-dependent. ECM stiffness-induced effects on junction assembly were dependent on the focal adhesion-associated F-actin linker and mechanotransducer talin and the TJ-regulated RhoA exchange factor GEF-H1. Similarly, defective morphogenesis could be rescued by inhibiting cytoskeletal contractility. Structural and functional studies further revealed that ZO-1-knockout cells under low tension assemble functional permeability barriers but with TJ that are structurally deficient. Our results thus indicate reciprocal regulation between cell mechanics and ZO-1, and identify tension-dependent and -independent roles of ZO-1 that regulate TJ formation.

## 2. Materials and Methods

### 2.1. Experimental Models

MDCK II cells were used for all experiments except the endothelial experiment shown in Appendix A. Three different sources of MDCK II cells were used that were kept separated to avoid differences originating from different substrains influencing the results. The RNA interference experiments targeting ZO-1 used the MDCK II strain we have described previously [23]. The GFP-mZO-1 MDCK cell line was generated using a mouse ZO-1 cDNA cloned into pEGFP-C1. Most experiments with MDCK II knockout cells were performed with cell lines described previously using two ZO-1 knockout clones (MDCK ZO-1KO C1 and C2), a ZO-1/ZO-2 double knockout clone (MDCK ZO-1/2KO), and the corresponding wild type MDCK II cell line (MDCK wt) [24]. None of the individual ZO-2 knockout clones tested showed sufficient depletion to be analysed further. A second set of knockout clones was used that had been generated in the Furuse laboratory and was hence marked with ‘Fur’ to distinguish the different MDCK knockout cell lines. These knockout clones were also previously described and used a Talen approach targeting the start ATG to generate the ZO-1 knockout clone but a CRISPR approach to knockout ZO-1 in the ZO-1/2 knockout; hence, the label for these samples were MDCKFur-wt, ZO-1KOFur, ZO-2KOFur, and ZO-1/2KOFur [17,25]. All cell lines were frozen as a contamination-free stock and fresh stock samples were put in culture every 2 months. Primary human dermal microvascular endothelial cells (C-12212) were purchased from PromoCell and cultured in 0.5% gelatin-coated dishes in in endothelial cell growth medium MV2 supplemented with C 39225 supplement mix (PromoCell) [11]. Cells were used between passage two and four. Cells were regularly stained with Hoechst dye to reveal nuclei and DNA of contaminants such as mycoplasma.

### 2.2. Experimental Setups

For experiments with a short culture time to monitor TJ formation during monolayer formation (Figures 1–6), cells were seeded at 3 × 10^5^ cells/well into 6-well plates and transfected with siRNAs the following day. After 24 h, the cells were trypsinised, resuspended in full medium, and seeded in experimental plates containing the required substrates. 7 × 10^3^ cells/well were used for 48-well plates containing glass coverslips uncoated or coated with Matrigel. 24-well plates containing 40 kPa or 1 kPa hydrogels were seeded with 12.5 × 10^3^ cells/well or 25 × 10^3^ cells/well, respectively. For immunoblotting, cells were plated accordingly using plates of different sizes with or without hydrogels. Experimental plates were processed after 48 h of culture. This plating concentrations were determined by titration and then chosen so that the cells reached comparable concentrations on all substrates and resulted in cultures consisting of cell islands and larger continents at the end of the experiments. Hence, cells reached only densities that still allowed them to spread if actomyosin concentration was increased of if they were plated on stiff substrates, enabling them to exert tension on cell–cell junctions. For experiments with knockout clones and short culture times, a corresponding two-step method was used to facilitate comparison (i.e., seeding at 3 × 10^5^ cells/well into 6-well plates for 48 h, before reseeding on different substrates at the same concentrations as for the RNA interference experiments. Experimental plates were also processed after 48 h of culture. For experiments with mature monolayers (Figures 7–11), 3 × 10^5^ cells were plated per well of a 24-well plate or in Transwell polycarbonate filters held in 24-well plates (0.4 μm pore size) and cultured for up to six days. Filter cultures were used for TER measurements using an EVOM and stick electrodes (World Precision Instruments) and for transmonolayer flux measurements with 4 kD FITC- and 70 kD Rhodamine-Dextran [26]. Fluorescence intensity was measured with a BMG FLUOstar OPTIMA microplate reader. The monolayers reached stable TER values after 3 days of culture.

### 2.3. Blebbistatin

Cells were treated with 10 µM Blebbistatin diluted from 10 mM stocks in DMSO overnight or for 4 h. Controls were incubated with a corresponding amount of DMSO. Details of the treatments for specific experiments are provided in the figure legends.

### 2.4. Plasmids

The canine E-cadherin tension sensor constructs (pTS-E-cadherin and pTS-E-cadherinΔCTD) were generously provided by Alexander R. Dunn (Stanford University, Stanford, CA, USA) [27] and a plasmid encoding mouse ZO-1 fused to GFP by Junichi Ikenouchi (Kyushu University, Fukuoka, Japan). The latter construct was used to generate pEGFP-mZO1 using the SacI and KpnI sites in pEGFP-C1. pEGFP-ZO-2 was a gift from Marius Sudol (Addgene plasmid # 27422; https://www.addgene.org/15974/ (accessed on 22 November 2022): 27422; RRID: Addgene_27422) [28]. pGW1-HA-ZO-2 was generously provided by Lorenza Gonzalez-Mariscal (Cinvestav, Mexico DF, Mexico) and Ronald Javier (Baylor College of Medicine, Houston, TX, USA) [29,30].

### 2.5. Transfections

siRNA transfections were performed using the RNAiMAX transfection reagent following the manufacturer’s instructions. Cells in 6-well plates were transfected by adding 8μL/well RNAiMAX and 8μL/well of 20 μM siRNA. For double knockdown experiments, siRNAs against two targets were mixed at a 1:1 ratio, and siRNAs against a single target were mixed at a 1:1 ratio with the control siRNA.The following canine siRNAs were used: canine ZO-1: 5′-CCAUAGUAAUUUCAGAUGU-3′ and 5′-CCAGAAUCUCGAAAAA GUGCC-3′; canine Talin: 5′-GUCCUCCAGCAGCAGUAUAA-3′ and 5′-GAGGCAACCACAGAACACAUA-3′; canine GEF-H1 5′-AGACACAGGACGAGGCUUA-3′, 5′-GGGAAAAGGAGAAGAUGAA-3′, and 5′-GUGCGGAGCGGAUGCGCGUAA-3′. Plasmid DNA was transfected using TransIT (Mirus Bio) by adding 1 µL of the reagent and 1 µg of DNA per 48-plate well. Transfections were left for 4 h before media replacement.

### 2.6. Antibodies and Immunological Methods

Immunological methods for immunofluorescence and immunoblotting were previously described [31,32,33]. Briefly, for immunofluorescence, cells were either fixed in 100% ice-cold MeOH for 5 min at −20 °C before being rehydrated in PBS for 5 min and incubated with blocking buffer (PBS 5% BSA, 10 mM glycine, 10% NaN_3_) for 30 min, or fixed in 3% PFA for 20 min at room temperature before permeabilization in PBS 0.3% Triton for 10 min and incubation in blocking buffer for 30 min. Cells were then incubated with primary antibodies overnight in blocking buffer at 4 °C, washed 3 times with blocking buffer, incubated with secondary antibodies diluted in blocking buffer at room temperature for 2 h, and washed 3 times with blocking buffer before mounting on glass slides with Prolong Gold (ThermoFisher Scientific. Waltham, MA, USA). For immunoblotting, cells were lysed in Laemmli sample buffer containing 6 M urea and freshly dissolved DTT at a final concentration of 0.3 M. The following commercial antibodies were used: occludin, mouse monoclonal (33-1500, ThermoFisher Scientific); ZO-1, mouse monoclonal (33-9100, ThermoFisher Scientific); ZO-2, mouse monoclonal (37-4700, ThermoFisher Scientific); Tricellulin, rabbit polyclonal (48-8400, ThermoFisher Scientific); Claudin-2, mouse monoclonal (32-5600, ThermoFisher Scientific); cingulin, mouse monoclonal (sc-365264, Santa Cruz Biotechnology, Dallas, TX, USA) and rabbit polyclonal (ab244406, abcam); claudin-1, rabbit polyclonal (51-9000, ThermoFisher Scientific); pMLC-S19, mouse monoclonal (3675, Cell Signaling Inc., Danvers, MA, USA) and rabbit polyclonal (3671, Cell Signaling Inc.); ppMLC-Thr18,S19, rabbit polyclonal (3674, Cell Signaling Inc.); p120-catenin, goat polyclonal (sc-373115, Santa Cruz Biotechnology, Dallas, TX, USA)); β-catenin, sheep polyclonal (ab65747, abcam); talin, mouse monoclonal (T3287, Sigma-Aldrich, Saint Louis, MS, USA); vinculin, mouse monoclonal (V9131, Sigma-Aldrich); and HA-epitope, rat monoclonal (11867431001, Roche, Basel, Switzerland). Rabbit polyclonal antibodies specific for ZO-2 and ZO-1, and mouse monoclonal and rabbit polyclonal antibodies against GEF-H1 were described previously [34]. The rabbit polyclonal antibody against claudin-4 was raised and affinity purified with a PRTDKPYSAKYSAAC peptide. Antibodies against JAM-A (rabbit polyclonal [35]; generously provided by Klaus Ebnet, University of Muenster, Muenster, Germany) α-tubulin (mouse monoclonal, [36], and podocalyxin/gp135 (3F2/D8, deposited to the DSHB by Ojakian, G.K. [37] were previously described. The following affinity-purified and cross-adsorbed secondary antibodies from Jackson ImmunoResearch were used: Alexa488-labelled donkey anti-rabbit (711-545-152) and anti-mouse (115-545-003), Cy3-labelled donkey anti rabbit (711-165-152) and anti-mouse (715-165-150) and Cy5-labelled donkey anti-goat (705-175-147) were diluted 1/300 from 50% glycerol stocks. Phalloidin-Atto647 (65906) and Hoechst-33342 were from Sigma-Aldrich. For immunoblotting, affinity-purified HRP-conjugated goat anti-mouse (115-035-003) and anti-rabbit (111-035-144) secondary antibodies for immunoblotting were purchased form Jackson ImmunoResearch and diluted 1/5000 from 50% glycerol stocks, and affinity-purified IRDye 800CW donkey anti-mouse (926-32212) and anti-rabbit (926-32213), and 680LT donkey anti-mouse (926-68072) and anti-rabbit (926-68023) secondary antibodies from LI-COR and diluted 1/10000. Immunoblotting was carried out using either enhanced chemiluminescence or fluorescence for detection as described [32,38]. Immunoblot images and were quantified with ImageJ/Fiji using the gel densitometry toolset.

### 2.7. Preparation of Matrigel-Coated Polyacrylamide (PAA) Hydrogels and Glass Coverslips

PAA hydrogel fabrication followed the same method as described in [9], which was adpated from the protocol used in [39]. Briefly, 22, 13, and 10 mm glass coverslips (Agar Scientific, Stansted, UK) were washed with 70% ethanol, dried at 70 °C, and exposed to UV light overnight. 22 and 10 mm coverslips were then incubated with a solution of 65μg/mL Matrigel (BD Biosciences, San Jose, CA, USA) for 2.5 h at 37 °C and 13 mm coverslips were silanized for 3 min using ethanol containing 0.37% Bind-Silane solution (GE Healthcare Life Science, Chalfont St Giles, UK) and 3.2% acetic acid. Mixes of PAA and bis-acrylamide (N,N′-methylenebisacrylamide, Sigma-Aldrich) corresponding to elasticity of 1 kPa and 40 kPa were prepared as described [40].

### 2.8. Preparation of PAA Hydrogels with Microbeads for Traction Force Microscopy

PAA hydrogels were prepared for TFM by adding 1 μL fluorescent carboxyl polystyrene beads (diameter = 0.20 μm; Dragon Green) to the PAA mix, before polymerisation. The E-modulus of the TFM gels ranged from 13.6 to 19.6 kPa (average of 16.4 ± 2.5 kPa). PAA/bis-AA mixes were sonicated and vortexed to avoid bead clumps and to ensure a homogenous distribution of the beads in the gels. We then added N,N,N′,N′-tetramethylethylenediamine (TEMED, Sigma-Aldrich) and 10% APS (Sigma-Aldrich) to the mixes at 0.1% and 0.8%, respectively, to prime polymerization. A 25 μL drop of activated mix was added on each coated 22 mm coverslips and silanized 13 mm coverslips were then quickly placed on top of the drops. Gels were let to polymerize for 30 min. The transfer of matrix proteins from the coated glass coverslip to the surface of the gels occurred by passive microcontact printing on PAA gels (μCP) [41]. Assemblies were then soaked in sterile H_2_O for 30 min. Gels were detached from the 22 mm coverslips using a scalpel blade and 13 mm coverslip-bound hydrogels were stored at 4 °C in sterile H_2_O containing 150 mM NaCl and 10 mM HEPES. Before seeding cells, gels were incubated with full MDCK culture medium for 30 min at 37 °C.

### 2.9. Generation of 3D MDCK Cysts

Generation of 3D epithelial cysts with MDCK cells was performed as described in [19]. Briefly, after trypsinization of siRNA-transfected MDCK cells, trypsin activity was quenched by diluting cells 1:3 in serum-containing medium. Cells were then resuspended to single cell suspensions and counted. Cells (20 × 103 in 5 μL) were mixed with 100 μL collagen-Matrigel master mix that was prepared as follows on ice: 60 μL of 1 mg/mL calf skin type I collagen (Sigma-Aldrich; C8919) was added to a buffering pre-mix containing 10μL 10× Dulbecco’s modified Eagle’s medium, 2 μL HEPES 1 M (pH 7.4) and 2 to 3 μL of 2 M NaOH. 10 μL 100% fetal bovine serum and 16 μL of Matrigel (growth factor reduced; BD Biosciences) were then added to obtain the final mix. Cells in the liquid matrices were plated in 48-well plates containing coverslips that had been previously incubated with 100 μL of the collagen-Matrigel master mix for 1 h at 37 °C. Matrices were allowed to solidify at 37 °C overnight before adding medium consisting of low glucose DMEM (Thermo Fischer) supplemented with 1% fetal bovine serum. Medium was replaced every 2 days. Spheroids were allowed to grow for 5 to 7 days before fixation with 3% paraformaldehyde for 20 min at room temperature followed by two washes with phosphate-buffered saline (PBS). Adding and aspirating solutions were done as slowly and gently as possible to avoid damaging the matrix. Cysts were then blocked and permeabilized for 30 min at room temperature in a blocking solution containing 2% BSA, 1% Triton X-100, and 25 mM Tris in PBS prior to staining with antibodies and mounting with Prolong Gold.

### 2.10. Light Microscopy

Immunofluorescence imaging of fixed cells was performed using a Nikon Eclipse Ti-E microscope with a CFI Apochromat Nano-Crystal 60× oil objective (N.A., 1.2) or a Leica TCS SP8 with an HC PL APO 40× (N.A., 1.30) or 63× (N.A., 1.40) oil objectives (z section thickness was 0.7 µm). Bright field imaging for 2D-morphogenesis assays on 1 kPa hydrogels was performed with a Super Plan Fluor 10× objective (N.A. 0.30). FRET imaging of the E-cadherin tension sensor was performed at 37 °C 20 to 24 h after transfection of the sensor plasmids using a Nikon Eclipse Ti-E inverted microscope equipped with excitation and emission CFP and YFP filters in external filter wheels with a CFI Apochromat Nano-Crystal 60× oil lens (N.A., 1.2) as described [9]. Crossover between CFP and YFP filters was calibrated by imaging individual fluorescent proteins expressed alone using all four emission/excitation filter combinations. FRET efficiency maps were then generated with the Nikon software with the built-in formula correcting for crossover between CFP and YFP channels. The Nikon Eclipse Ti-E microscope was also used to perform static TFM imaging [9]. Acquisitions of bright field images of cell islands and images of fluorescent beads were done using a 20× objective. Focus was maintained using the Nikon Perfect Focus system. 1 × 103 to 3 × 103 cells transfected with siRNAs were seeded on hydrogels two days before imaging. This range of seeding concentrations were optimized to obtain well separated islands of 10–20 cells on the day of imaging; hence allowing to obtain a sufficient margin of empty surrounding matrix to have sufficiently large no-displacement fields in the images. Cell islands were imaged first in bright field before image stacks of the beads embedded in the contracted substrate were taken in the FITC channel. A solution of 1 M NaOH + 1% sodium dodecyl sulphate was used to lyse cells before acquiring stacks of beads in the relaxed state.

### 2.11. Freeze Fracture Analysis and Electron Microscopy

Freeze fracture electron microscopy was done as previously described [42]. In brief, the MDCK clones were fixed with phosphate-buffered glutaraldehyde (2%) and then subsequently incubated in 10% (*v*/*v*) and then in 30% (*v*/*v*) glycerol. After freezing them in liquid nitrogen-cooled Freon 22, cells were fractured at −100 °C and then shadowed with platinum and carbon in a vacuum evaporator (Denton DV-502). Resulting replicas were treated with sodium hypochlorite, picked up on grids (Ted Pella Inc., Redding, CA, USA), and analysed with a video-equipped Zeiss 902 electron microscope (Carl Zeiss AG, Jena, Germany; Olympus item Veleta).

### 2.12. Image Processing and Quantifications

Microscopy and immunoblot images were processed and adjusted using ImageJ/Fiji and Adobe Photoshop CC software. Panels were created using Adobe Photoshop CC software. For quantification of FRET images, ImageJ/Fiji was used to calculate mean integrated densities of junctional segments above measured backgrounds [9]. For the quantification of confocal xz-sections and tricellular junction accumulation, mean integrated densities of ROIs manually drawn around detectable junctions were calculated and normalized by the one representing the total cell area.

TFM experiments in the present study were done in parallel of the ones described in [9] and hence, share the same methods. Experimental drift before and after cell lysis was corrected using the ImageJ Template Matching plugin from Dr. Qingzong Tseng (https://sites.google.com/site/qingzongtseng/template-matching-ij-plugin; accessed on 22 November 2022). A Matlab script developed by Martial Balland [43,44] was used to perform all computations including particle image velocimetry (PIV), beads tracking, displacement field calculations (linear interpolation on a regular grid with 2.6 μm spacing), unconstrained Fourier Transform Traction Cytometry, traction map generation and strain energy calculations. Islands with an out of equilibrium force over 10% of the sum of the individual force amplitudes were discarded. A final filtering based on island areas was done to obtain comparable pools of island between conditions. Finally, we obtained the strain energy densities by dividing the total strain energy U [45] calculated from the traction and displacement fields by the xy area covered by cell islands. We then averaged absolute values of single islands across individual gels and expressed these averages as folds of the control condition.

Nuclei counting for discriminating dense and sparse areas was performed with images of Hoechst-stained nuclei that were automatically counted in xy images using a pipeline of custom ImageJ/Fiji macros which used a Cellpose pretrained model for nuclei segmentation [46]. Acquisition with the Nikon Eclipse Ti-E microscope using the 60× objective generated fields of view with an area of 1.6 × 10^4^ μm^2^. Fields of view of cells seeded on glass were considered as sparse when nuclei count was between 10 and 25, and dense when nuclei count was over 40. On 40 kPa hydrogels fields were considered sparse when nuclei count was between 15 and 30 and considered dense when it was over 50. No discrimination was done for cells grown on 1 kPa hydrogels as they did form densely packed islands in all experimental conditions.

Discrimination of basal from apical ppMLC staining features in z-stacks acquired on the Nikon Eclipse Ti-E microscope using the 60× oil objective was performed by selecting substacks of slices most in focus at basal and lateral sides of the cells, before projection using the “extended depth of focus” plugin from the CLIJ2 library [47]. Grayscale images were then processed using the Image Differentials plugin [48] from the ImageJ/Fiji BIG-EPFL update site, which applies a differentiation operation on images seen as continuous functions. The Smallest Hessian operation was chosen to enhance in-focus features while attenuating out-of-focus ones.

Basal F-actin enhancement was performed on z-stacks of phalloidin-labelled cells obtained with the Nikon Eclipse Ti-E microscope with the 60× oil objective and F-actin signal enhancement was computed in ImageJ/Fiji. Briefly, the most basal in-focus slice in each z-stack was manually selected and projected using the “extended depth of focus” plugin from the CLIJ2 library [47]. Projections were then convolved using a homemade kernel. Then, after subtracting the background, a Gaussian blur and a Median filter were applied. Finally, the brightness and contrast were adjusted.

F-actin distribution in xz vertical cross-sections was quantified from xz-section images obtained with the Leica TCS SP8 microscope with the 40× oil objective. Apical and basal F-actin staining were manually outlined in single cells and the ratio of the mean fluorescence intensity was calculated for each cell.

Quantification of the junctional formation index for occludin and p120-catenin was performed using multichannel z-stack images acquired on the on the Nikon Eclipse Ti-E microscope using the 60× oil objective. The analysis of junction formation was done using a semi-automatic pipeline of macros in ImageJ/Fiji. Briefly, most in focus slices for each channel were selected using the “Find focused slices” plugin developed by Qingzong Tseng (https://sites.google.com/site/qingzongtseng/find-focus; accessed on 22 November 2022) and stacks were projected in each channel using the “extended depth of focus” plugin from the CLIJ2 library [47]. Channels were merged back into a composite image and junctions were manually traced using the Line Tool along the visible signal or based on other junctional markers when the analysed junctional marker was not forming a junction. Tracing was done along the mediatrix between the centres of two nuclei when no clear junction could be seen. Two other lines were traced on both sides along the junctions within the cytosol. The mean fluorescence intensity of the junctional line was divided by the average of the mean fluorescence intensities of the two cytosolic lines to obtain the junctional formation index. The line width used for the analysis of occludin staining was 15px, and 20px for p120-catenin staining.

General features of the distribution of ppMLC in immunostained cells grown on glass, 40 kPa and 1 kPa hydrogels were analysed in z-stacks obtained with the Nikon Eclipse Ti-E microscope using the 60× oil objective. Most in-focus slices were manually selected to discriminate lateral from basal features. Active myosin bundling at cell periphery and sharp lateral active myosin lines at cell junctions were manually counted. The percentage of cells displaying each feature was calculated per image after having obtained the nuclei count as described above.

The morphogenesis index quantification of cell islands grown on 1 kPa hydrogels was performed using bright-field xy images acquired as described above and automatically analysed using a custom macro in ImageJ/Fiji. Briefly, island outlines were obtained by applying the built-in “Find edges” plugin before applying image cleaning processes (“Despeckle”, “Dilate”, “Fill holes”). Cleaning was further completed with an opening operation from the morphological filters of David Legland’s MorpholibJ plugin [49]. The built-in “Analyze particles” plugin was applied on the final binary images to obtain the number of cell island per image, reflecting the monolayer fragmentation, and the area of each island, to estimate the coverage of the monolayer per image. The cumulated area of islands per image was expressed as the percentage of coverage, and the ratio of the former to the number of islands was defined as the morphogenesis index.

3D epithelial cysts were generated and imaged as described above. Multichannel z-stacks were carefully visualized along the z-axis to discriminate different distinct lumens in the spheroids when multiple lumens were visible. The number of lumens per cyst was manually counted and the counts of cysts having defined amounts of lumens per cysts were plotted to compare the distribution of the cyst populations between conditions.

Morphological features of the focal adhesions were analysed by automatically processing immunofluorescent staining of vinculin with a custom macro in ImageJ/Fiji. Briefly, the images were pre-processed using the built-in “Subtract Background...” plugin before binarization with the “Adaptative Threshold” plugin developed by Qingzong Tseng (https://sites.google.com/site/qingzongtseng/adaptivethreshold; accessed on 22 November 2022). The “Watershed Irregular Features” plugin from Jan Brocher’s BioVoxxel Toolbox (“BioVoxxel” update site) was then applied. Single focal adhesions areas and maximum Feret diameters were then obtained using the built-in “Analyze particles” plugin. Data were expressed as the average of the measurements per image.

Cell morphometric analysis from multichannel z-stack images of cells stained for junctional markers or F-actin was performed following the same method as described in [9]. It consists of a pipeline of custom ImageJ/Fiji macros. Briefly, a binary mask of cell–cell junctions was produced by using the Ridge Detection plugin [50] on maximum projections of the analysed staining. Small cytosolic objects were cleared using the Analyze Particle tool. The mask obtained was then overlaid on a multichannel RGB images with slices picked manually for the best focus for each channel, and cell edges were corrected based on AJ (p120-catenin or α-catenin staining) and manually traced if no TJ was detectable. Cell segmentation was then produced using the Tissue Analyzer plugin [51], which produced ROIs corresponding to single cells and random colour maps of the segmentations. Morphometric data were computed directly from the ROIs after exporting them back into Fiji/ImageJ.

The morphometrical analysis of TJ ultrastructure was performed using images taken with a video-equipped Zeiss 902 electron microscope (Carl Zeiss AG, Jena, Germany; Olympus iTEM Veleta) at a magnification of 51,000×. Within the images, vertical grid lines were drawn at 200 nm intervals perpendicular to the most apical TJ strand and the horizontally oriented number of strands within the main TJ meshwork was counted at intersections with the grid lines. Meshwork depth was measured as the distance between the most apical and contra-apical strand, and strand discontinuities within the main compact TJ meshwork of >20 nm were defined as “breaks”. Strand appearance was noted as “particle type” or “continuous type”. For the wt cells 53 TJs were analysed, while for the ZO-1 KO 28 and 25 TJs were analysed. As for the ZO-1/2 KO only rarely TJ with strands were detected; in addition to these, areas where TJ clearly would be present were analysed, leading to 29 images that were used.

### 2.13. Statistics and Reproducibility

For the quantifications shown, provided data points refer to the numbers of cells, TJ, tricellular junctions, image fields or cell islands analysed per type of sample (this information is provided in figure legends) and are derived from at least three independent experiments. All immunoblots represent repeats of 3 or more experiments. Statistical significance was tested using nonparametric Kruskal–Wallis and Wilcoxon tests. The FRET experiments are shown as standard box plots (25th to 75th percentiles, with a line indicating the median, and the analysed numbers of cells are provided in the figure legends). All other quantifications show all data points along with the mean and standard deviation for each category, as well as the median and interquartile ranges. For the statistical analysis of the TJ ultrastructure, Student’s t-test with correction for multiple testing (Bonferroni-Holm) was used and for the proportions of strand appearance, the Z-test also with Bonferroni-Holm adjustment was performed. *p*-values below *p* = 0.05 were defined as statistically significant. Graphs and statistical calculations were generated with JMP-Pro or GraphPad Prism.

## 3. Results

### 3.1. Matrix Stiffness Modulates ZO-1 Regulation of Cell Morphodynamics

ECM stiffness, which regulates actomyosin function and cell spreading via focal adhesions [1,4], determines the mechanical load on ZO-1 in MDCK cells [9]. It is thus possible that ZO-1′s role in TJ assembly may depend on the stiffness of the ECM. Therefore, we asked if and how ZO-1 depletion regulates morphogenetic processes on different substrates. ZO-1 was efficiently depleted with two distinct siRNAs (Figure 1A–C). Depletion by the pooled siRNAs was kept as the main condition to limit off-target effects. To control specificity, a GFP-tagged mouse ZO-1 expressing cell line was used (GFP-mZO-1). Although GFP-mZO-1 expression was lower in these cells than endogenous ZO-1, the level was not affected by transfection of siRNAs against canine ZO-1 (Figure 1A,B).

To analyse cell and monolayer morphogenesis, experimental parameters were adjusted to analyse cells during the initial steps of 2D epithelial morphogenesis during which dynamics is high (i.e., migrative edge of islands, continent formation) and cells can exert high tensile forces on adhesion complexes, but with cell islands large enough to discriminate densely packed centres from sparser edges [52]. Hence, siRNA-transfected MDCK cells were plated for 48 h after seeding at cell densities too low to form fully confluent monolayers.

Figure 1D–F shows that ZO-1 knockdown led to a dramatic morphological change when seeded on glass. The cells were flatter, more elongated (~3-fold increase in xy area) and had an apparently weaker lateral F-actin belt but increased basal stress fibres (Figure 1D–F and Appendix A). The phenotype was rescued by GFP-mZO-1 expression (Figure 1D and Appendix A). As expected, control MDCK cells appeared larger on stiff than on soft ECM because stiff ECM stimulates actomyosin activity and cell spreading (Figure 1D–F and Appendix A) [4,53]. ZO-1 depletion also induced increased spreading on 40 kPa hydrogels (~1.5-fold increase), but without the strong induction of basal stress fibres observed on glass. ZO-1-depleted cells grown on the soft 1 kPa hydrogels appeared slightly smaller than control cells. ZO-1 localisation was not affected by the substrate stiffness as expected from previous observations of epithelial cells grown on soft and stiff ECM (Appendix A) [9,54]. Hence, ECM stiffness determines the impact of ZO-1 depletion on epithelial morphodynamics.

**Figure 1 cells-11-03775-f001:**
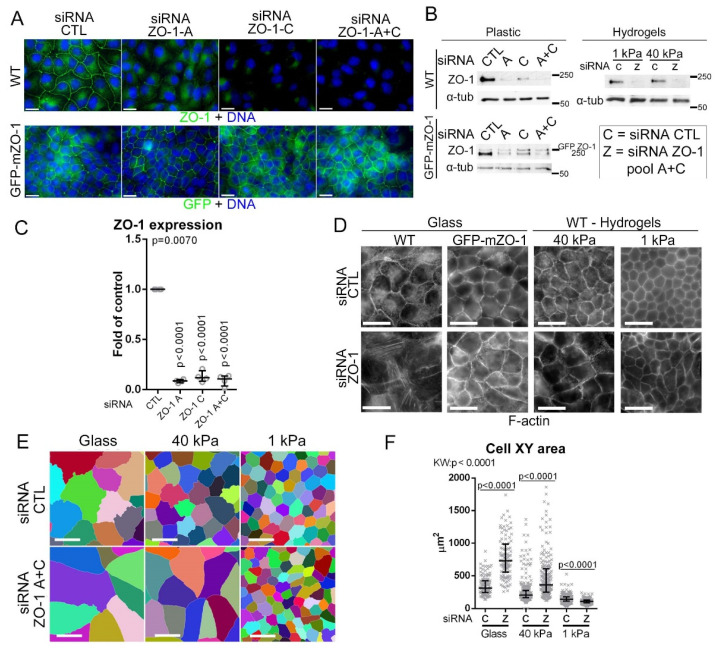
ZO-1 regulation of cell morphogenesis is ECM stiffness-dependant. (**A**–**C**) siRNA-mediated depletion of ZO-1 in control and mouse GFP-tagged ZO-1 (GFP-mZO-1)-expressing MDCK cells using two different siRNAs against ZO-1. Cells were analysed by immunofluorescence ((**A**), upper row, ZO-1; lower row, GFP; nuclei were labelled with Hoechst) or immunoblotting ((**B**,**C**); α-tubulin was blotted as a loading control). ZO-1 downregulation was quantified by densitometry ((**C**), data points are derived from independent determinations). (**D**–**F**) siRNA transfected wild-type and GFP-ZO-1 cells were seeded on ECM of different elasticities. Cells were stained for F-actin (**D**) and 2D cell segmentation was performed based on junctional immunostaining using the method described by Haas et al. (see also Materials and Methods) (**E**,**F**) to quantify single-cell XY areas (data points show individual cells analysed) [9]. Magnification bars, 20 µm.

### 3.2. ZO-1 Regulates Junction Formation and Myosin-II Activity and Distribution

Staining of the TJ protein occludin revealed that ZO-1 depletion led to disrupted junctions in cells grown on glass, with a twice as strong effect in sparse areas (Figure 2A). On 40 kPa hydrogels, the junctional disruption was less pronounced in dense areas, but remained high in sparse ones. Conversely, we observed no TJ disruption in cells grown on 1 kPa or in the GFP-mZO-1 cell line grown on glass (Figure 2A and Appendix A). Hence, substrate stiffness determines the importance of ZO-1 for TJ formation.

ZO-1 regulates myosin activation in different cell types and conditions [11,16]. Treating cells with the myosin-II inhibitor blebbistatin rescued TJ formation in ZO-1 depleted cells grown on glass or 40 kPa; hence, actomyosin activity is required to inhibit TJ formation (Figure 2A,B and Appendix A). Stimulation of actomyosin activity by ZO-1 depletion was validated by immunoblotting, revealing increased double-phosphorylated myosin light chain (ppMLC) relative to control cells, regardless of matrix stiffness (Figure 2C–E). The distribution of ppMLC was strongly affected by the ECM with increased peripheral ppMLC bundles in ZO-1 depleted cells grown on glass (especially in sparse cells), which was more modest in cells grown on 40 kPa (Figure 2F and Appendix A). No differences were observed in ppMLC localisation in cells grown on 1 kPa upon ZO-1 depletion. A striking feature of ppMLC distribution on glass, which was not detected on hydrogels, was the appearance of lateral sharp filaments at cell–cell junctions (Figure 2F and Appendix A). Blebbistatin treatment counteracted formation of those peripheral structures, indicating that their formation requires myosin activity (Figure 2H and Appendix A). Staining for the AJ marker p120-catenin did not reveal any clear signs of AJ disruption upon ZO-1 depletion (Figure 2I).

**Figure 2 cells-11-03775-f002:**
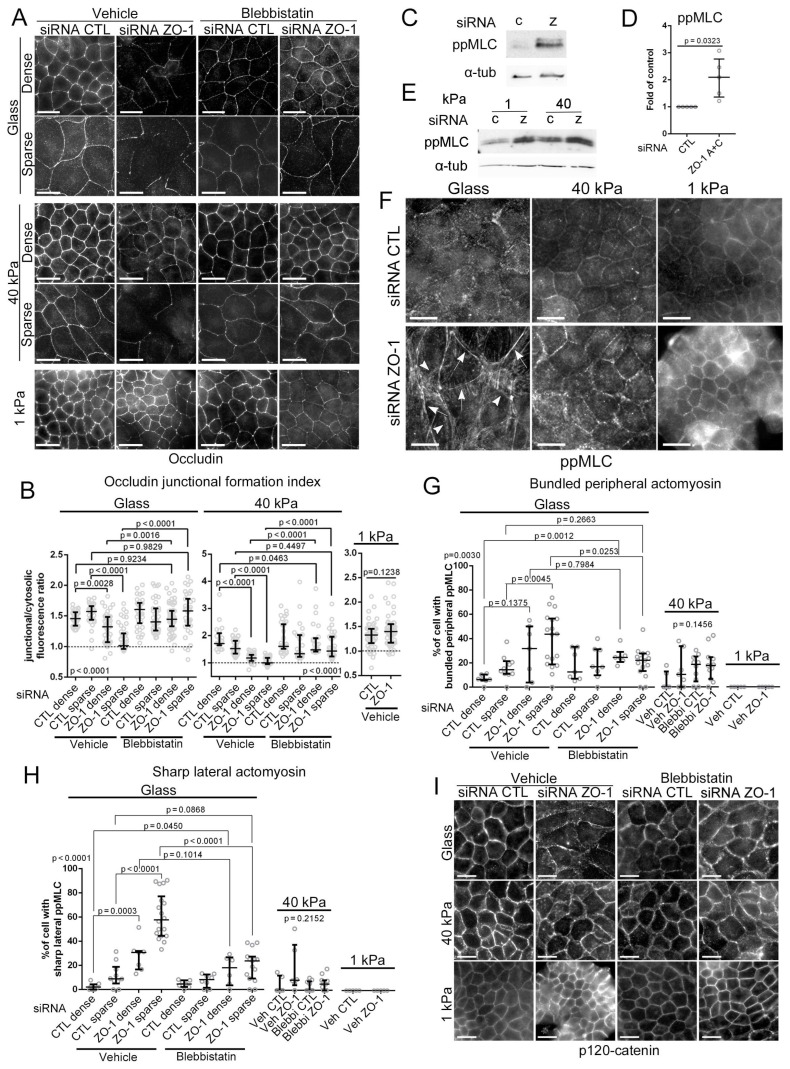
ZO-1 is required for junction formation on stiff ECM. (**A**,**B**) TJ formation by siRNA-transfected MDCK cells plated on ECM of different stiffnesses was analysed by immunofluorescent staining of occludin. Cells were either treated with DMSO (vehicle) or blebbistatin for 24 h. Quantification of junction formation was performed by measuring single-junction ratios of junctional to cytosolic occludin fluorescence intensity number of cells in each condition (**B**). (**C**–**E**) Actomyosin activity was evaluated by immunoblotting for ppMLC and α-tubulin as a loading control. Panel (**D**) shows quantification of the ppMLC levels by densitometry of cells plated on glass (data points are independent determinations). (**F**–**H**) Immunofluorescence of ppMLC was also performed in cells grown on glass or hydrogels to analyse the distribution of active myosin within cells. The percentage of cells per image displaying thick peripheral ppMLC bundles ((**G**); labelled in panel (**F**) with arrowheads) and sharp lateral ppMLC ((**H**); labelled in panel (**F**) with arrows) were manually quantified considering areas with densely and sparsely populated cell areas (n, number of images in each condition). (**I**) Cells were stained for the AJ marker p120-catenin. Magnification bars, 20 µm.

Depletion of ZO-1 thus disrupted TJ, but not AJ, formation in an ECM stiffness-dependant manner and led to a stiffness-dependant reorganisation of active myosin. The morphological phenotypes could be rescued by blebbistatin treatment, indicating a mechanistic link between ZO-1 control of junction formation and actomyosin activity and organisation.

### 3.3. ZO-1 Controls 2D Morphogenesis on Soft ECM

Reduction of ZO-1 expression in MDCK cells using shRNAs disrupts morphogenesis in 3D Matrigel ECM [19,55]. We observed a comparable phenotype upon depletion of ZO-1 with siRNAs (Appendix A). The 3D morphogenesis defect was attributed to the absence of a spatial cue as those cell lines formed morphologically normal monolayers in 2D. However, 3D Matrigel gels are very soft (<1 kPA) [56]; hence, it is also possible that low ECM stiffness combined with deregulation of actomyosin activity causes the morphogenesis defect. Therefore, to eliminate effects caused by the absence of a spatial cue, we tested whether ZO-1 depletion affects monolayer morphogenesis on soft ECM in 2D.

While ZO-1 depleted cells had intact cell–cell junctions on 1 kPa ECM (Figure 2), they formed only small islands that failed to form large islands of continuous monolayers (Figure 3A,B). Monolayers formed normally on 40 kPa hydrogels or glass. Monolayer formation was rescued by treating cells with blebbistatin or by mouse ZO-1 expression (Figure 3A,B). Control GFP-mZO-1 cells displayed continuous but smaller islands compared to wild-type cells, reflecting reduced proliferation upon ZO-1 transfection [57].

To validate this phenotype, we analysed CRISPR/Cas9-generated knockout MDCK clones [24]. Two distinct ZO-1 knockout clones were used along with a clone lacking expression of both ZO-1 and ZO-2 (Figure 3C). All knockout clones displayed defective 2D morphogenesis on 1 kPA hydrogels (Figure 3D–G). No significant differences were observed between single and double knockout clones, indicating that ZO-1 is required for epithelial morphogenesis on soft but not stiff ECM. Therefore, the role of ZO-1 in epithelial morphogenesis depends on ECM stiffness.

### 3.4. Regulation of Mechanical Tension at AJ by ZO-1 Is Dependent on Matrix Stiffness

The contraction of ZO-1 depleted cells on soft ECM could be stimulated by increased cell–cell tension. Depletion of ZO-1 indeed affects monolayer and AJ tension in different experimental model systems. However, the reported effects are contradictory. While in primary human endothelial cells and zebrafish embryos ZO-1 depletion/knockout led to reduced cell–cell tension [11,12], depletion of ZO proteins in MDCK cells was reported to stimulate increased mechanical tension at least on a stiff substrate [58]. The reason for these differences are not clear.

To measure whether ZO-1 depletion stimulates increased cytoskeletal tension on cell–cell AJ, we transfected MDCK cells grown on different substrates with an E-cadherin FRET tension sensor to analyse the effect of RNAi-mediated ZO-1 depletion on tension acting on AJ [27]. ZO-1 knockdown led to increased tension at AJ when cells were grown on Matrigel-coated glass as indicated by the decrease in lateral FRET efficiency (Figure 4A–C). In contrast, in cells on Matrigel-coated elastic ECM, ZO-1 depletion led to reduced AJ tension. Thus, substrate stiffness determines the impact of ZO-1 depletion on tension acting on AJ. These data also indicate that the ZO-1 depletion-induced morphogenesis defect on 1 kPa ECM was not due to increased cell–cell tension.

**Figure 3 cells-11-03775-f003:**
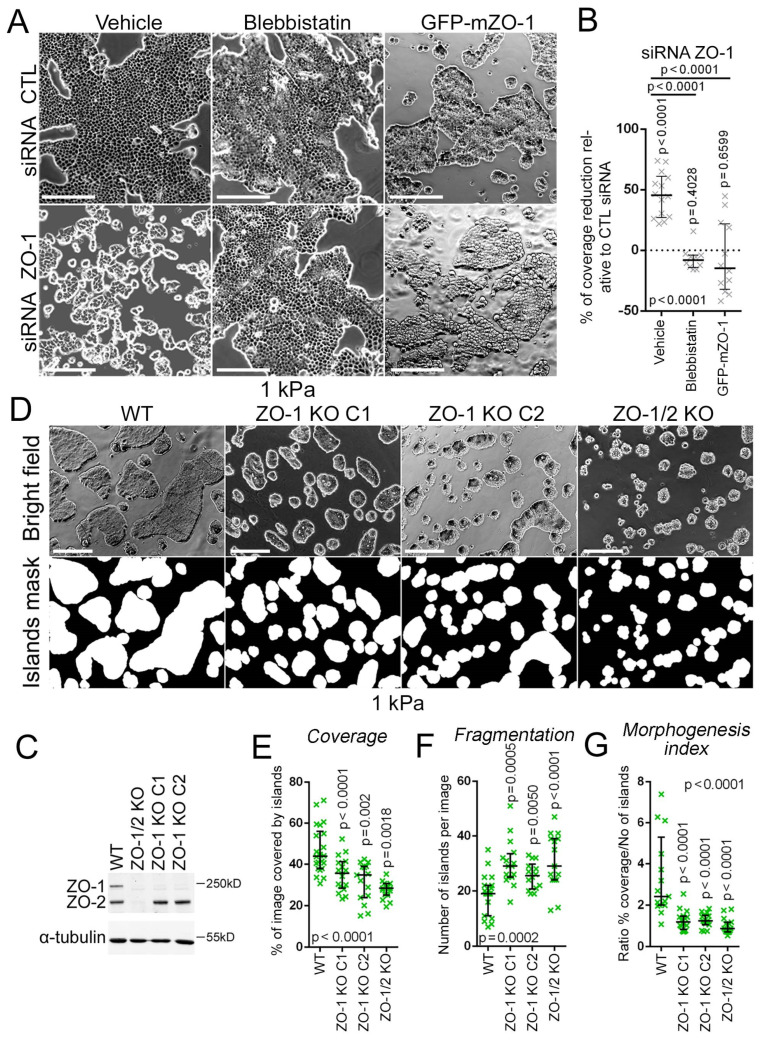
ZO-1 is required for epithelial morphogenesis on soft ECM. (**A**,**B**) siRNA-transfected MDCK cells were seeded on 1kPa hydrogels and treated with DMSO (vehicle) or blebbistatin for 24 h before bight-field microscopy imaging ((**A**), left and middle panel). GFP-ZO-1 expressing cells were also transfected with siRNA and seeded on 1 kPa hydrogels before imaging ((**A**), right panel). Percentages of area coverage of cell islands was measured ((**B**), data points represent images). (**C**–**G**) Wild-type and knockout MDCK cells were analysed by immunoblotting (**C**) or were seeded on 1 kPa hydrogels before bright-field microscopy imaging and binary segmentation of cell islands (**D**). Area coverage by cell islands per image (**E**), the number of islands per image as a measure of fragmentation (**F**), and the morphogenesis index consisting of the ratio of both parameters (**F**) were quantified. Data points represent images. Magnification bars, 200 µm.

To determine possible differences between MDCK cells grown on different substrates, we stained for F-actin and observed that, unlike cells grown on glass, cells grown on soft ECM of 40 or 1 kPa hydrogels displayed strong apical and lateral F-actin and little basal F-actin (Figure 4D,E). Staining for double phosphorylated myosin regulatory light chain as a measure for active myosin-II, revealed strong lateral staining. Substrate stiffness thus regulates actomyosin organisation in MDCK cells. Intriguingly, endothelial cells, which respond with reduced cell–cell tension to ZO-1 depletion [11], exhibit a similar actomyosin organisation as MDCK cells on soft ECM (Appendix A). Hence, the effect of ZO-1 on the tension acting on E-cadherin correlates with the organisation of F-actin as in cells that have a preferential junctional actin organisation ZO-1 depletion leads to reduced tension on E-cadherin whereas in cells with a strong basal cytoskeleton ZO-1 depletion stimulates tension on E-cadherin. The correlation between F-actin organisation and impact of ZO-1 depletion on cell–cell tension supports a model in which the cytoskeletal architecture of different cell types determines the effect of ZO-1 depletion on cell–cell tension.

**Figure 4 cells-11-03775-f004:**
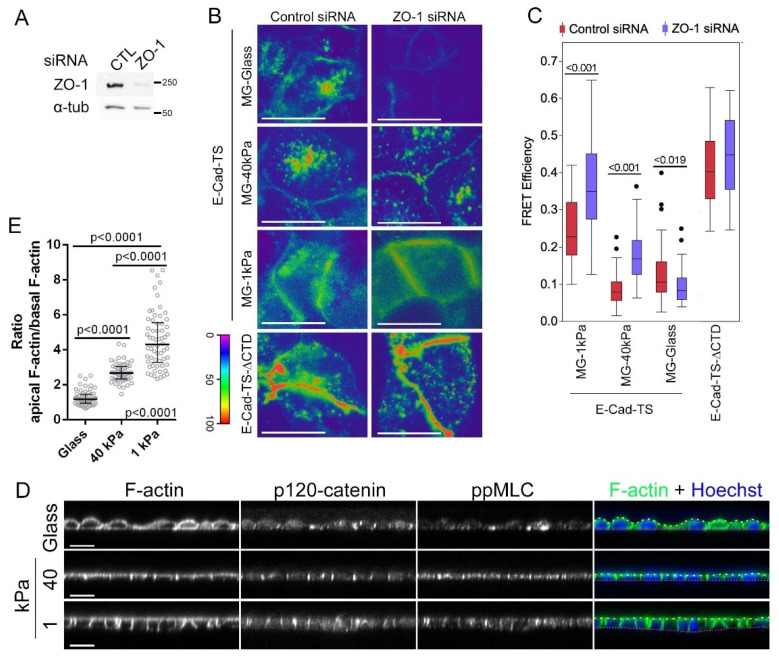
ECM stiffness guides ZO-1′s function in regulation of AJ tension. (**A**) Immunoblots confirming ZO-1 depletion by transfection of siRNAs targeting ZO-1 but not by non-targeting (CTL) siRNAs. α-tubulin is shown as a loading control. (**B**,**C**) Cells transfected with siRNAs as in panel (**C**) were seeded on substrates of different stiffnesses coated with Matrigel (MG) before transfection with an E-cadherin tension sensor constructs with E-cad-TS or without the C-terminal cytoplasmic domain of E-cadherin. FRET analysis was performed by epifluorescence microscopy (**D**). Panel (**E**) shows FRET efficiencies at cell–cell contacts (n is the number of cells analysed for control and ZO-1 siRNAs: E-Cadherin-TS MG-Glass, 46 and 45; MG-40 kPa, 59 and 59; MG-1 kPa 40 kPa, 47 and 48; E-Cadherin-TS-ΔCTD, 28 and 29). (**D**,**E**) MDCK cells seeded on glass or hydrogels of 40 or 1 kPa stiffness were fixed and stained for F-actin, p120-catenin and ppMLC. Shown are confocal z-sections (**D**). The right-hand column shows overlays of F-actin in green with Hoechst in blue in which the apical side of the monolayers have been traced with yellow and the basolateral side with red dashed lines. Single cell ratios of apical over basal F-actin fluorescence intensities were calculated ((**E**); data points represent values from individual cells). Magnification bars, (**A**), 20 μm; (**D**), 10 µm.

### 3.5. ZO-1 Regulates Cell Traction on the ECM

The increased actomyosin activity induced by ZO-1 depletion could stimulate increased contractility along the basal membrane and, hence, leading to the observed cell shape changes and possibly morphogenesis defects. Therefore, we next asked if ZO-1 regulates focal adhesion function. Immunostaining of vinculin in ZO-1 depleted cells seeded on glass revealed that focal adhesions were strongly remodelled upon ZO-1 depletion, appearing thicker and more elongated (Figure 5A–C). Such changes are signs of increased mechanical load on basal adhesions [59]. Therefore, we next used traction force microscopy to measure how ZO-1 depletion impacted the forces exerted by cells on the substrate.

Figure 5D–F shows that depletion of ZO-1 in cells seeded on stiff polyacrylamide hydrogels resulted in increased strain energy densities of more than 50%. Hence, ZO-1 controls the traction forces cells exert on the substrate, raising the possibility that myosin-dependent junction disruption on stiff ECM are due to pulling forces generated when cells are grown on substrates that offer sufficient mechanical resistance. Therefore, we tested whether interfering with the cytoskeletal linkage and mechanotransduction at focal adhesions can rescue TJ formation by depleting talin, which links focal adhesions to the F-actin cytoskeleton [3], either on its own or together with ZO-1. Immunoblotting showed that double knockdown resulted in comparable depletion levels to the ones obtained with individual siRNA transfections (Figure 5G). We performed analogous experiments depleting GEF-H1 as this RhoA activator is inhibited by intact TJ and regulates the response to mechanical forces on integrins (Figure 5H) [60,61]. Immunofluorescence of occludin revealed that depletion of both talin or GEF-H1 in ZO-1 knockdown cells rescued TJ formation (Figure 5I,J). Talin did not have obvious effects on myosin activity along the basal membrane, whereas GEF-H1 depletion appeared to lead to reduced pMLC staining along the basal part of the cells (Appendix A). However, the changing cell sizes made clear conclusions difficult. Talin depletion also led to a compaction of cell islands reminiscent of cells grown on 1 kPa ECM, supporting the conclusion that ECM stiffness and traction generated at focal adhesions are central to the ZO-1 depletion phenotype.

Thus, ZO-1 acts as a regulator of cell-ECM traction and both reduction of ECM stiffness or inhibition of focal adhesion function counteract the effect of ZO-1 depletion on junction formation, indicating that the main ZO-1-specific role during junction formation is control of cytoskeletal tension and the resulting forces acting on cell–cell and cell-matrix adhesion complexes.

### 3.6. The ZO-1 Knockout Enhances the Depletion Phenotype

We next analysed the functional role of ZO proteins in junction formation in knockout clones as RNAi-mediated depletion may be affected by variable silencing efficiencies. When cells were seeded as described for the knockdown experiments, ZO-1 knockout clones displayed defective TJ formation, comparable to ZO-1 knockdown cells (Figure 6A,B). The ZO-1/2 double knockout lacked junctional occludin staining as expected [24,25]. Although AJ appeared intact, p120-catenin staining intensity at junctions was reduced with increased cytosolic staining, a feature we had not observed in the ZO-1KD model (Figure 6A,C); hence, the more efficient removal of the protein in the knockout led to an enhanced phenotype.

Seeding the knockout cells on elastic ECM again led to a rescue of TJ formation and the 1 kPa ECM did so more efficiently than the stiffer one (Figure 6D). However, the occludin distribution remained discontinuous in both ZO-1KO clones analysed even on 1kPa ECM. No rescue of TJ formation occurred with the double knockout cells, which only formed apparently contracted islands as observed in Figure 3D. ZO-1 knockout stimulated MLC phosphorylation and a striking redistribution of F-actin and active myosin, with increased stress fibres and bundles of lateral phosphorylated MLC in cells plated on glass (Figure 6E and Appendix A). However, in the double knockout cells, actomyosin bundles remained more basally distributed.

**Figure 5 cells-11-03775-f005:**
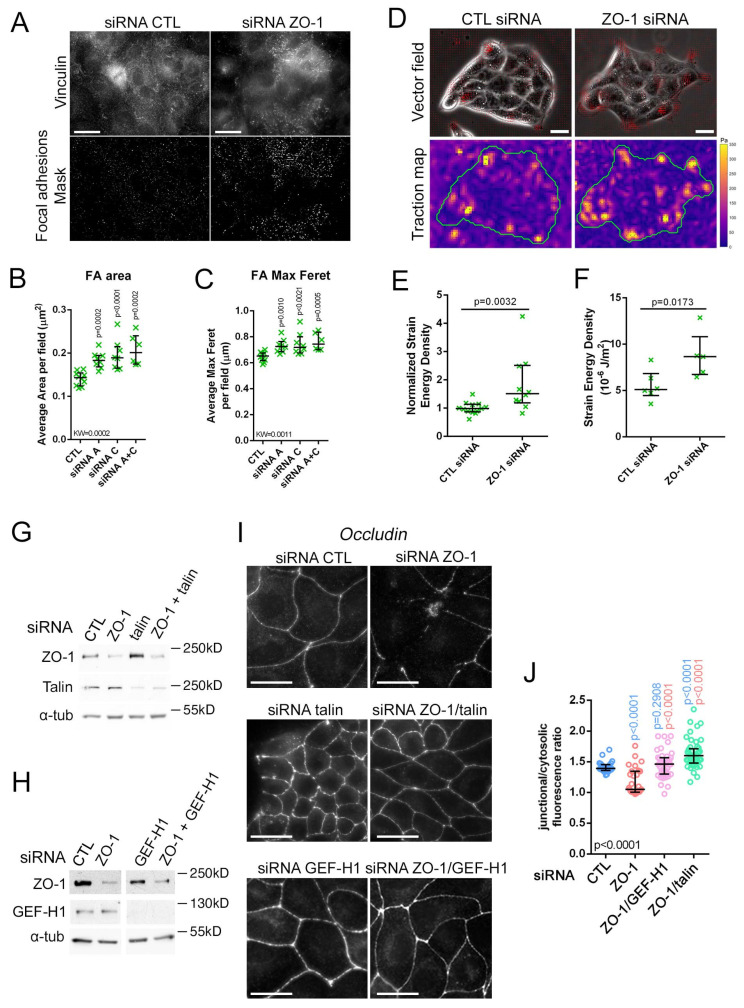
ZO-1 regulates basal adhesion architecture and cell traction on the ECM. (**A**–**C**) siRNA-transfected MDCK cells were immunostained for vinculin to observe focal adhesions by imaging the most basal z-slices of the cells ((**A**), upper images). Binary masks of focal adhesions were computed from the vinculin immunofluorescence images ((**A**), bottom images). From the masks, morphological parameters were analysed: adhesion areas (**B**) and the maximal Feret diameters as a measure of elongation (data points represent cells analysed). (**D**–**F**) TFM performed on siRNA-transfected MDCK cell islands. Traction vector fields were overlaid on phase contrast images ((**D**), top) and the corresponding stress maps ((**D**), bottom) were computed. Average strain energy densities of all islands in a single gel were normalized to respective controls (**E**) number of gels in each condition and absolute values of single islands of one representative experiment were plotted number of islands in each condition (**F**). The effect of ZO-1 depletion was measured in parallel with that of JAM-A depletion [9]; hence, the same control siRNA values are shown. (**G**–**J**) MDCK cells transfected with siRNAs as indicated (the total concentration of siRNA was kept constant by adding control siRNA) were plated on glass and were then analysed by immunoblotting (**G**,**H**) or immunofluorescent staining of occludin (**I**,**J**). The junctional relative to cytosolic specific fluorescence intensity was quantified as a measure for junction assembly ((**J**), data points represent analysed cells, blue p values are for comparisons to control and red values for comparisons to ZO-1 siRNA only). Magnification bars, 20 µm.

**Figure 6 cells-11-03775-f006:**
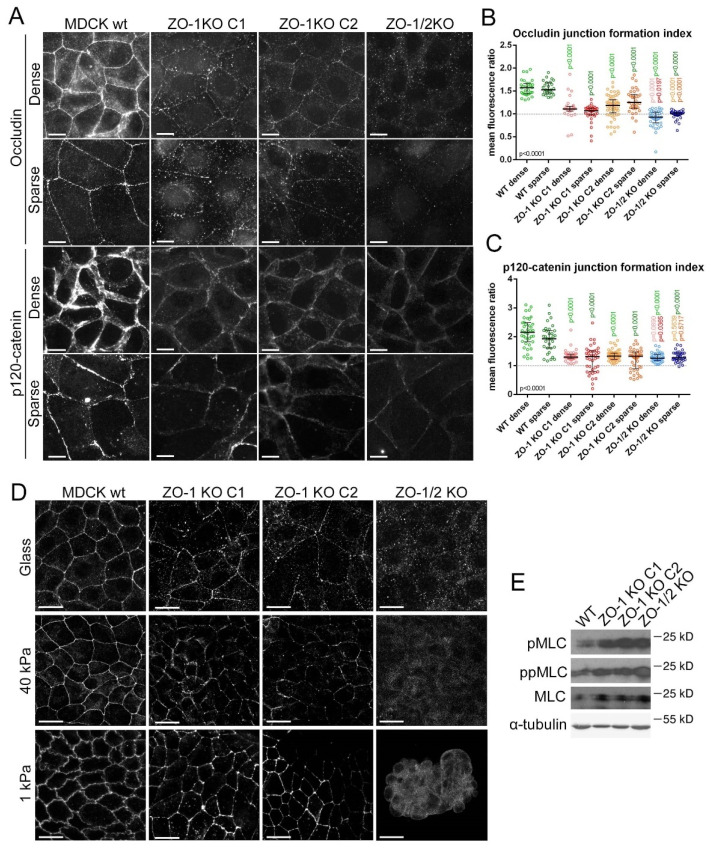
Non-redundant role of ZO-1 in TJ formation. (**A**–**C**) Wild-type and knockout MDCK cells were seeded on glass for 2 days as Figure 3 and immunostained for occludin and p120-catenin (**A**). Quantification of the ratio of junctional to cytosolic mean fluorescence intensity for single junctions was performed for occludin (**B**) and p120-catenin (**C**) on images derived from dense and sparse areas (data points represent junctions analysed). (**D**) Wild-type and knockout MDCK cells were cultured until full confluence before immunostaining for occludin. Note, the reduced junctional occludin staining in ZO-1 knockouts and the discontinuities on all ECM types. On 1kPA ECM, the double knockout cells were unable to form a continuous confluent monolayer. (**E**) Wild-type and knockout MDCK cells were analysed by immunoblotting for phosphorylated MLC. Magnification bars, 20 µm.

ZO-1 knockout and siRNA-mediated depletion thus led to fundamentally similar phenotypes with a TJ formation defect that was dependent on ECM stiffness. However, the knockout phenotype was stronger, possibly reflecting the residual ZO-1 expression in the knockdown cells.

### 3.7. ZO-1 and ZO-2 Have Different Functions in TJ Formation

The occludin staining in the ZO-1 knockout clones appeared discontinuous in the cultures plated on 1 kPa ECM (Figure 6D); hence, we tested whether formation of a continuous occludin distribution required more time in ZO-1 knockout cells. However, even after longer time of culture, occludin appeared in patches aligned with cell–cell contacts (Figure 7A). The cytosolic junctional proteins ZO-2, cingulin, ZO-3 and GEF-H1 displayed strongly increased cytosolic pools and, if junctional staining was detected, it was patchy (Figure 7B and Appendix A). The discontinuous junctional occludin staining was not affected by blebbistatin, indicating that it was not due to increased actomyosin activity (Figure 7C). Hence, neither soft ECM nor direct myosin inhibition could rescue the discontinuous distribution of occludin, indicating that the phenotype was not caused by increased actomyosin activity in ZO-1KO cells but was cytoskeletal tension-independent. Staining for p120-catenin was comparable with wild-type cells in all clones, suggesting that the AJ phenotype observed at lower density reflected delayed AJ formation.

To determine if ZO-2 expression is required for normal TJ formation, we expressed GFP-tagged ZO-1 or ZO-2 in double knockout cells. Expression of GFP-ZO-1 efficiently recovered TJ formation (Figure 8A and Appendix A). In contrast, much of the expressed GFP-ZO-2 remained cytosolic and the pool that was at junctions formed patches. Occludin followed the junctional ZO-2 pattern and cingulin remained primarily cytosolic. Hence, GFP-ZO-2 was unable to rescue normal TJ assembly.

We next tested the role of endogenous ZO-2 by employing recently described individual ZO-2 knockout cells along with a TALEN-generated ZO-1 knockout and a corresponding double knockout cell line [17,25]. The TALEN ZO-1 knockout had not revealed such a strong effect on other junctional proteins or junctional patches of occludin as we observed here in ZO-1 knockout clones [17]. Immunoblotting revealed that ZO-2 depletion was efficient; however, ZO-1 was still expressed at low levels in the TALEN-based ZO-1 knockout (Figure 8B). The remaining expression may be due to an in-frame ATG shortly after the TALEN-mutated translational start codon (the ZO-1 allele in the double knockout was generated using a CRISPR-Cas9 approach [25]). Both, the partial ZO-1 knockout as well as the ZO-2 knockout still supported a continuous occludin distribution (Figure 8C). As the knockouts had been generated in a different strain of MDCK cells, we also re-expressed ZO-1 and ZO-2 in double knockout cells and observed that ZO-1, but not ZO-2, was efficiently recruited to cell junctions and that only ZO-1 induced a continuous linear junctional distribution of occludin (Figure 8D). Thus, ZO-1 and ZO-2 have distinct functions and differ in their potential to induce TJ formation as ZO-1 was required for normal TJ formation, but ZO-2 was dispensable.

**Figure 7 cells-11-03775-f007:**
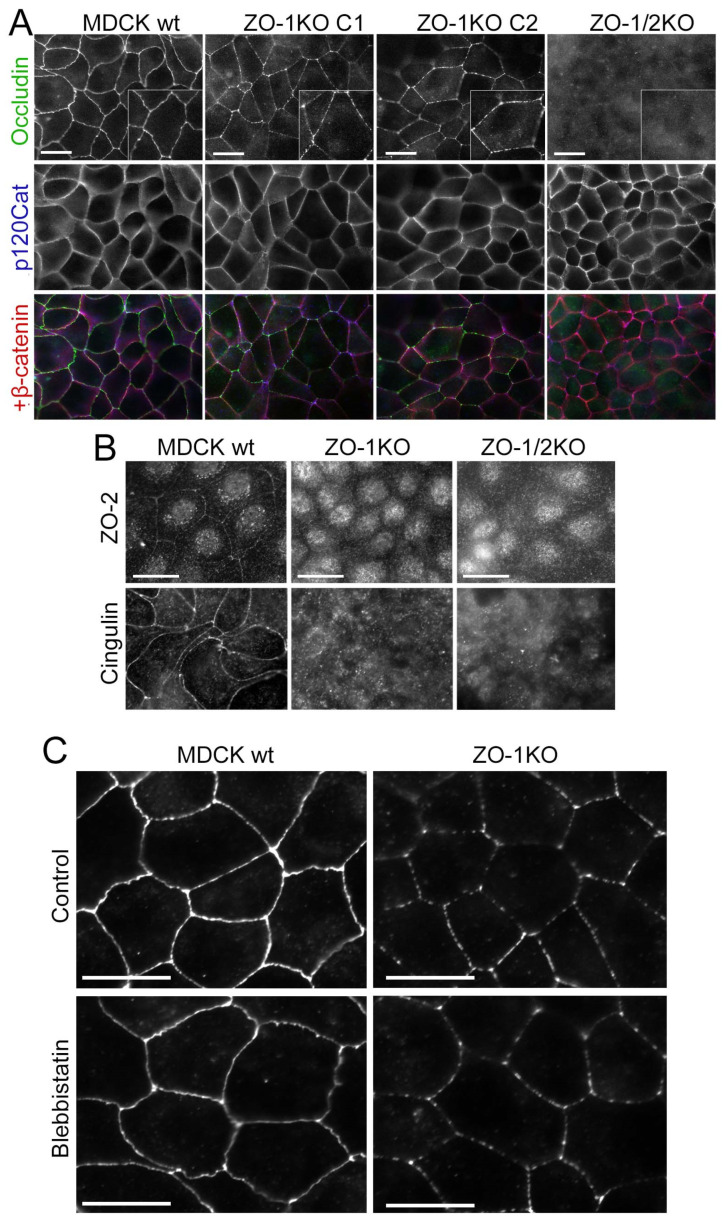
ZO-1 and ZO-2 have distinct functions in TJ assembly. Analysis of wild-type and knockout MDCK cells seeded on glass and grown to confluence for 5 days by epifluorescence microscopy. Insets show higher magnifications to illustrate the discontinuous staining of occludin (**A**,**B**) Cells were stained for the junctional proteins. Note, occludin remained discontinuously distributed and recruitment of cytosolic TJ proteins was strongly attenuated. (**C**) Wild-type and ZO-1KO MDCK cells were treated with blebbistatin and then fixed and stained for occludin. Note, the distribution of occludin remained discontinuous in blebbistatin-treated ZO-1KO cells, indicating that the disrupted occludin arrangement was not due to the increased myosin activity. Magnification bars, 20 µm.

**Figure 8 cells-11-03775-f008:**
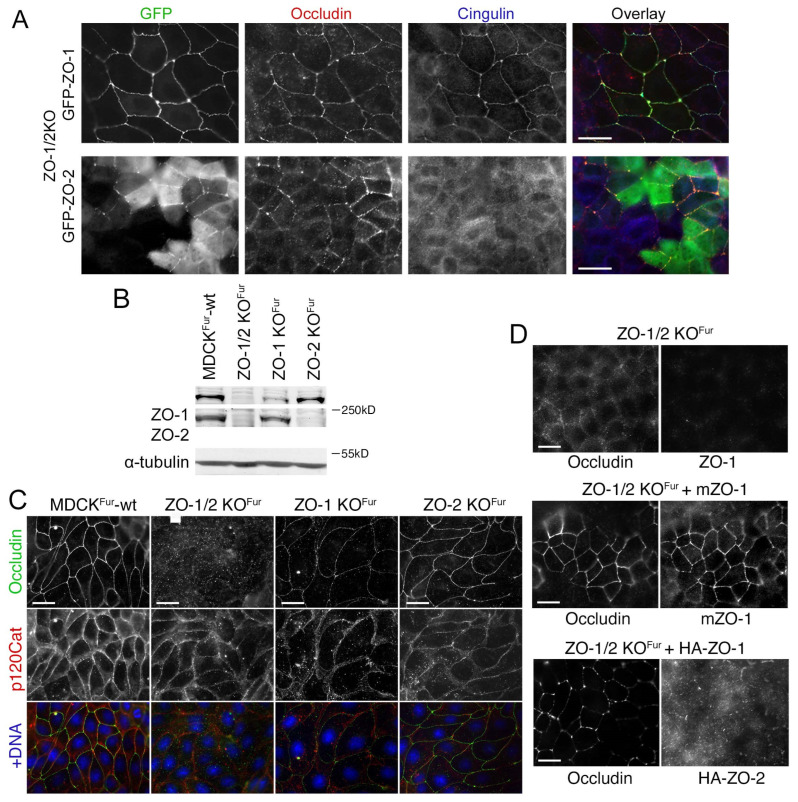
ZO-1 and ZO-2 have distinct functions in TJ assembly. (**A**) Double knockout cells were transfected with GFP-ZO-1 or GFP-ZO-2 and TJ formation was analysed by immunostaining occludin and cingulin. Note, only GFP-ZO-1 induced normal distribution of TJ proteins (see Appendix A for expression of GFP-ZO-2 in wild-type MDCK cells. (**B**,**C**) Wild-type and knockout MDCKFur cells were analysed by immunoblotting to monitor expression of ZO-1 and ZO-2 (**A**, antibodies against epitopes in the C-terminal domain of the ZO proteins was used as in Figure 4C) or immunofluorescence to analyse junction formation (**B**). (**D**) ZO-1 and ZO-2 were re-expressed in ZO1/2 knockout MDCKFur cells followed by staining for occludin to monitor TJ formation. Magnification bars, 20 µm.

### 3.8. ZO-1 Knockout Increases Transepithelial Permeability and Disrupts TJ Ultrastructure

The defects in TJ assembly observed in ZO-1 knockout cells at low tension prompted the question how junctional permeability was affected. To test this, we plated the cells on filter inserts at high confluence. We first tested whether mature monolayers on filters still exhibited the junctional patches of membrane proteins. Figure 9A–D and Appendix A show that junctional recruitment of occludin and different claudins was reduced and their junctional distribution irregular in ZO-1 knockout cells with the most striking discontinuities displayed by occludin. The recruitment of the cytosolic protein cingulin was disrupted (S8D). The least affected protein was JAM-A; however, like other junctional membrane proteins, its TJ recruitment was strongly reduced in ZO-1 knockout cells (Figure 9D and Appendix A). Confocal z-sectioning also indicated irregularities in the arrangement of the nuclei as they often appeared arranged in an irregular manner in cells lacking ZO-1 rather than localising close to the basal end of the cells, suggesting that ZO-1 regulates nuclear positioning (Figure 9D,E).

**Figure 9 cells-11-03775-f009:**
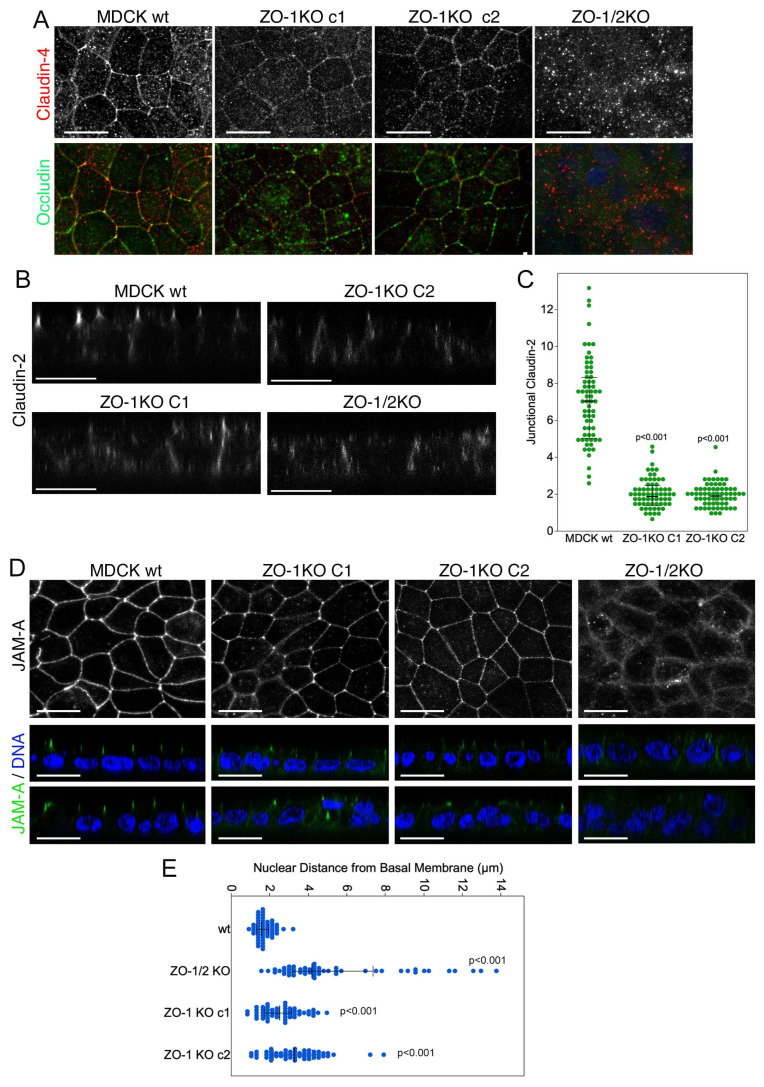
ZO-1-deficiency inhibits junctional protein recruitment in filter-grown cells. Confluent wild-type and knockout MDCK seeded on permeable supports were analysed by immunofluorescence confocal microscopy (**A**,**B**,**D**). All confocal XY sections are maximum intensity projections of all sections containing junctional staining. Additional junctional markers and xy-sections along with a quantification of z-sections of samples stained for JAM-A and occludin are shown in Appendix A. Panel (**C**) shows quantification of junctional claudin-2 staining on z-scans. Panel (**E**) shows a quantification of the nuclear position derived from z scans of JAM-A samples analysed in (**D**). Data points in panels (**C**,**D**) are derived from individual cells. Magnification bars, 20 µm.

We next analysed the barrier function by measuring transepithelial electrical resistance (TER) and paracellular tracer flux of 4 and 70kD fluorescent dextran. TER was ~10% lower in the two ZO-1 knockout clones once the monolayer reached an equilibrium (Figure 10A and Appendix A). No significant increase in permeability for the 70kD tracer was observed, indicating no general defects in monolayer integrity (Figure 10B). However, 4kD dextran diffusion was increased by ~4-fold, indicating increased size-selective diffusion. Thus, ZO-1 knockouts were still able to form functional barriers despite the discontinuous distribution of junctional membrane proteins; however, both ion conductance and small tracer permeability increased. Neither TER nor increased permeability were affected by myosin inhibition, further supporting the conclusion that the phenotype observed in high density monolayer was cytoskeletal tension-independent (Figure 10C,D).

**Figure 10 cells-11-03775-f010:**
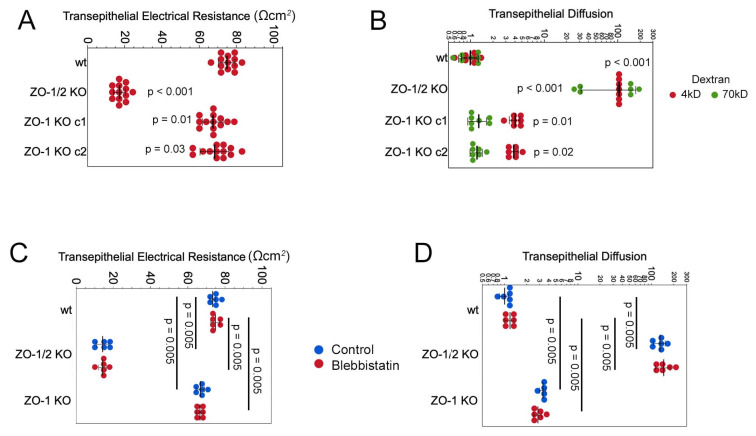
ZO-1-deficiency deregulates monolayer permeability. (**A**,**B**) TER and paracellular flux assays with fluorescent dextrans of different molecular weights were performed with monolayers of the same age as those analysed by immunofluorescence (**C**,**D**) TER and paracellular flux assays were repeated as in panels A and B in the presence of the myosin inhibitor blebbistatin. Data points in all panels represent individual filter cultures.

TJ contain a network of intramembrane strands that can be detected by freeze fracture-electron microscopy [8,62]. These strands are thought to be formed by claudin and occludin family proteins. Freeze fracture electron microscopy revealed that in wild type cells strands predominantly appeared as particle-type as expected [63]. This was largely also the case for the ZO-1 knockout clones; however, their number was halved (Figure 11). The meshwork extension was also reduced (Figure 11B). In the ZO-1/2 KO cells, not only was the presence of TJ strands uncommon, those that were detected showed a strongly reduced number of strands and meshwork extension (Figure 11A,B) and a tendency towards a more continuous appearance. In the ZO-1 KO clones, strands did not seem disrupted as in the double knockout clone but, due to the reduced number, we often just detected single strands; in the ZO-1/2 KO even the strand-like areas appeared unorganized (Figure 11C). In addition, there was a tendency of increased breaks and structural defects in tricellular corners. Although clear tricellular corners were only rarely found and could hence not be evaluated quantitatively, they appeared to consist only of the central tube and lacked the typical branches (Figure 11D). Tricellulin, a transmembrane protein required for strand branching in tricellular junctions [64], was strongly reduced at tricellular junctions (Figure 11E,F). Thus, while the discontinuous junctional distribution of membrane proteins was paralleled by striking changes in TJ ultrastructure, the ZO-1 knockout cells were still able to form intramembrane strands that appeared largely morphologically normal.

**Figure 11 cells-11-03775-f011:**
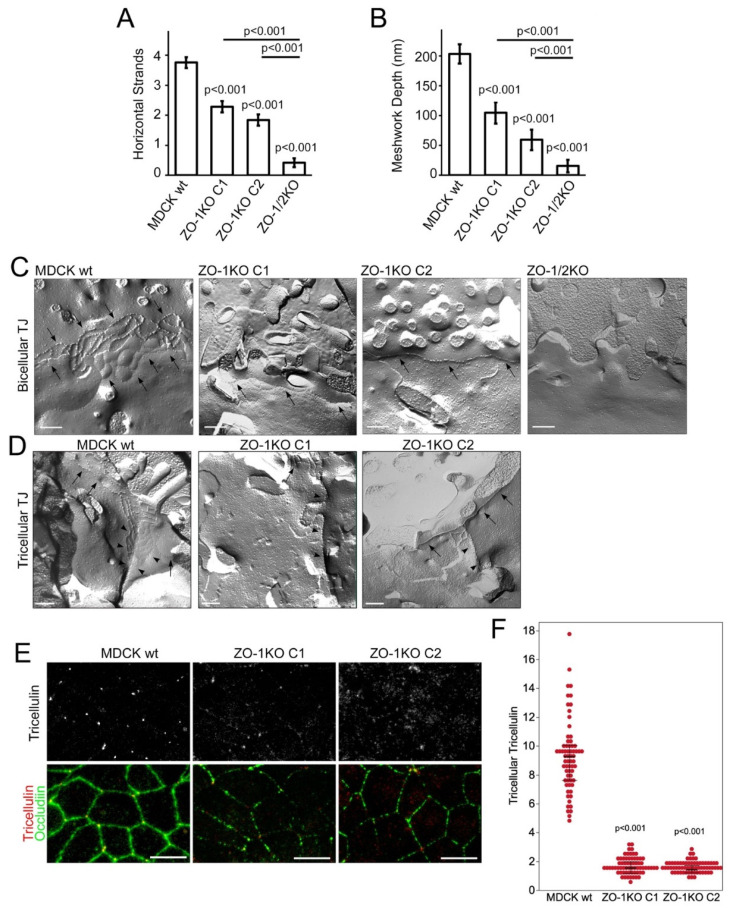
ZO-1-deficiency disrupts TJ ultrastructure. Freeze fracture analysis of TJ ultrastructure of confluent control (wt) and knockout MDCK cells. (**A**) Numbers of horizontal strands were reduced in ZO-1 KO cells as well as in the ZO-1/2 double KO clone. The double KO cells exhibited even more strongly reduced numbers of strands compared to the ZO-1 KO cells. Note, often no or only one TJ strand could be observed in various replicas of the double KO. (**B**) The meshwork extension was drastically reduced in ZO-1 KO cells as well as in the ZO-1/2 KO cells. Note, for occurrence of one strand no extension existed. The quantifications are based on the analysis of MDCK wt, 53; ZO-1KO C1, 28; ZO-1KO C2, 25; and ZO-1/2KO, 29 TJ. (**C**) Representative images of bicellular TJ in MDCK wt and the respective ZO-1 and ZO-1/2 KO cells. Arrows indicate intramembrane strands. (**D**) Images of tricellular TJ of MDCK wt and the ZO-1 KO cells suggested that in the ZO-1 KO only the central tube was present and the typical horizontal branches as seen in the wild-type controls cells were missing. Arrowheads point to intramembrane strands that are part of tricellular junctions and arrows to strands in bicellular junctions. (**E**,**F**) Confocal maximum intensity projections of junctional xy-sections taken from cells stained for tricellulin and occludin. Panel (**F**) shows quantifications of accumulation of tricellulin in tricellular corners. Magnification bars, (**C**,**D**) 200 nm; (**E**) 20 µm.

## 4. Discussion

Our results indicate that ZO-1′s function in TJ assembly and cell morphogenesis depends on the mechanical properties of the cell, and, in turn, that ZO-1 regulates the forces acting on adhesion complexes mediating interactions with neighbouring cells and the ECM. While functional TJ can form without ZO-1 in monolayers plated on very soft ECM, TJ remain structurally deficient, indicating that the role of ZO-1 in junction assembly extends beyond transmission and regulation of cell–cell tension. Hence, ZO-1 has tension-dependent and -independent functions that regulate TJ assembly and epithelial morphogenesis (Figure 12).

**Figure 12 cells-11-03775-f012:**
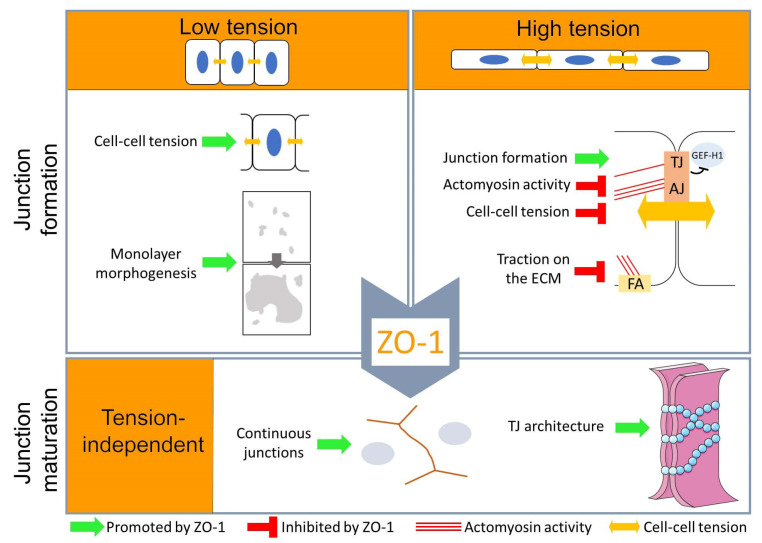
Schematic of the functions of ZO-1 during TJ assembly and epithelial morphogenesis. On soft ECM, ZO-1 is not required for assembly of functional barriers but regulates epithelial morphogenesis and cell–cell tension by tuning contraction and, thereby, epithelial sheet integrity. On stiff ECM, ZO-1 regulates contractility and ECM traction to control TJ assembly. In a second, tension-independent step, ZO-1 is required to assemble morphologically and structurally fully assembled TJ; in its absence, TJ remain only partially assembled and functionally altered.

ZO-1 carries a mechanical load and is thought to play a central function in mechanosensing at TJ [9,12]. ZO-1 regulates tension on AJ and overall cell–cell tension; however, the results reported are contradictory. While in primary endothelial cells in vitro and zebrafish embryo in vivo cell–cell tension decreased upon ZO-1 depletion, MDCK monolayers depleted of both ZO-1 and its close homologue ZO-2 exhibited increased cell–cell tension [11,12,58]. Our data now provide a possible explanation for the different observations reported. The effect of ZO-1 depletion on AJ tension in MDCK cells depended on ECM stiffness, which also determined actin cytoskeleton architecture. ZO-1 depletion on stiff ECM led to a stronger induction of stress fibres and actomyosin fibres at the apical periphery, likely to promote monolayer tension. Hence, the physical properties of the ECM determine how ZO-1 impacts cell–cell tension.

Primary endothelial cells, which display reduced AJ tension upon ZO-1 depletion [11], exhibit a similar actomyosin organisation to MDCK cells on soft ECM. Similarly in zebrafish, the cell–cell contacts shown to have reduced tension in response to ZO-1 knockout are actomyosin-enriched [12]. Therefore, these observations support a model in which the relative enrichment of the lateral cytoskeleton compared to the basal membrane determines the role of ZO-1 in cell–cell tension regulation. Hence, in cell sheets in which tension is primarily determined by junctional forces ZO-1 positively impacts AJ tension, and in monolayers in which tension is determined by high ECM stiffness ZO-1 negatively regulates tension by inhibiting basal actomyosin activity.

ZO-1 not only regulates cell–cell tension but also traction forces acting on the ECM and, thereby cell and monolayer morphology. The assay employed to measure ECM traction requires the use of a soft ECM (~16.4 kPa); hence, increased traction forces are not limited to very stiff matrices. On very soft matrices (1 kPa), cells with reduced ZO-1 expression failed to form large islands of monolayers but contracted, which was actomyosin-dependent. This disrupted 2D morphogenesis phenotype correlates with results obtained with 3D cultures, which require very soft ECM (<1 kPA) [56]), showing that depletion of ZO-1 or double knockout of ZO-1 and ZO-2 disrupts epithelial cyst morphogenesis [19,25,55].

Cells on 1 kPa ECM also exhibited increased myosin activation upon ZO-1 depletion but, in contrast to cells on stiffer ECM, this did not lead to breaks in junction formation, suggesting that the induced monolayer contraction was sufficient to dissipate increased cytoskeletal tension. This is in agreement with the well-established principle that stiff ECM promotes cell spreading [3,4]. Thus, ZO-1 depletion modulates both mechanical forces acting on AJ as well as on the ECM; and depletion of ZO-1 inhibits TJ formation only if ECM is sufficiently stiff to support cell spreading and, hence, a high mechanical load on cell–cell junctions. ZO-1 is directly exposed to such ECM regulated tension as ECM stiffness regulates the mechanical load on ZO-1 [9]. The role of ECM stiffness-driven cell spreading is further supported by the consequences of inhibiting focal adhesion function by talin depletion that, as a very soft matrix, inhibited cell spreading and rescues TJ formation upon ZO-1 depletion.

Our data thus indicate that ZO-1 depletion leads to cell-wide changes in cell mechanics that are modulated by the overall actin organisation of the cell. This is supported by the recent observation that ZO proteins regulate tissue fluidity in migrating cells [65]. It is thus likely that the biological importance of ZO-1 and its impact on morphogenetic processes are dependent on other factors that regulate organisation of the actin cytoskeleton, such as the cell type or ECM properties. It will be interesting to study such roles of ZO-1 in other dynamic biological processes. For example, ZO-1 knockout in mice leads to defective notochord and neural tube organisation at E9.5 [18]. These tissues represent the stiffest area of the developing embryo around this developmental stage [22]. Hence, mouse embryonic development may rely on regulation of cell and tissue mechanics by ZO-1.

We found that ZO-1 has specific tension-dependent and -independent functions that regulate TJ assembly. In cells on stiff ECM, ZO-1 was required for efficient TJ formation, which was actomyosin-dependent. ZO-1 depletion is well established to stimulate myosin-II activation [11,16] and did so also on 1 kPa ECM, on which TJ formation was not ZO-1 dependent. However, on 1 kPa hydrogels junctions are under low tension (Figure 4; ref. [9]; hence, ZO-1 was only required for TJ formation if junctions had to support high tension. A tension-dependent role for ZO-1 in TJ formation is compatible with previous studies on the role of ZO-1 in calcium-switch experiments on glass [15] or in primary endothelial cells [11]. The requirement of ZO-1 for TJ formation in cells under high tension is further supported by the observations that depletion of the focal adhesion mechanotransducer talin or the RhoA exchange factor GEF-H1, which stimulates cell spreading and stress fibre formation, rescued TJ formation. GEF-H1 is inhibited by TJ recruitment [60] and remained cytosolic in ZO-1 deficient MDCK cells. Hence, one of the functions of ZO-1 during TJ formation is the coordination of junction formation with GEF-H1 recruitment and inactivation.

ZO-1 interacts with both transmembrane proteins and the cytoskeleton [10] and carries a mechanical load that is dependent on actomyosin activity [9]. The tension acting on ZO-1, like TJ formation in its absence, is dependent on ECM stiffness, suggesting a model in which ZO-1 supports junction formation by adapting the cytoskeletal forces generated in the cell body to the forming TJ. However, such a cytoskeletal linker function may already be important prior to integration of ZO-1 into the forming junction. TJ formation involves a phase separation of cytosolic junctional proteins that is driven by ZO proteins [12,24]. Phase separation occurs prior to arrival of ZO-1 at TJs, and movement of the condensates to the junction is powered by actomyosin activity [12,66]. However, alternative roles of ZO-1 are likely to contribute to the tension-dependent effects, such as its role in the regulation of cell signalling and, in particular, the junctional recruitment and spatial regulation of RhoA exchange factors [8].

ZO-1 also serves a tension-independent role at TJ (Figure 12). In monolayers under low tension, TJ assembled but junctional proteins were very inefficiently recruited and distributed irregularly along the junction. While reduced TJ protein recruitment had previously been observed in ZO-1 knockout cells, the effect was modest and no discontinuities of junctional membrane proteins were observed, which was likely due to incomplete depletion [17]. Given the low remaining levels of ZO-1 expression in those knockout cells, only small amounts of ZO-1 are required to mediate the continuous linear junctional arrangement of other TJ proteins. This is also a likely reason why the effects of ZO-1 depletion were missed in previous studies using MDCK cells carrying shRNA plasmids [16]. ZO-1 was also knocked out in other cell types, and no effects on TJ were reported [20,21,67]. However, those knockout alleles involved removing early exons (part of PDZ1); hence, residual expression of truncated forms or genetic compensation mechanisms may have prevented detection of the phenotype [68].

The biochemical and functional properties causing the differences between ZO-1 and ZO-2 is not clear. However, ZO-1 is more efficiently recruited and more stably associated with TJ [24]. It is possible that regulatory mechanisms that regulate phase separation during TJ formation may be the underlying reason as the three ZO proteins differ in their propensity for phase separation [24] or that ZO-1 recruits signalling proteins that control junction assembly and stability [69].

ZO-1 knockout also disrupted the normal ultrastructure of TJ. TJ strand formation is reduced at bi- and tricellular corners in cells lacking TJ transmembrane proteins [25,64,70]. Hence, attenuation of strand formation as indicated by reductions in the number of horizontal strands and meshwork depth was in line with the reduced recruitment of TJ components. As identification of TJ in fracture replicas is biased towards areas that contain strands, it cannot be excluded that strand numbers are even lower than measured. Nevertheless, the local presence of morphologically detectable intramembrane strands is compatible with the patchy distributions of junctional membrane proteins detected by confocal microscopy since the latter covered a large junctional area, and inducing a discontinuous distribution of an individual junctional protein like occludin does not affect the appearance of intramembrane strands [31]. The presence of TJ strands in the ZO-1 knockout cells is certainly in agreement with the remaining TER in those cells. The reduced number of strands could also explain the increase in selective paracellular permeability as diffusion would be expected to occur more quickly across a TJ with fewer strands [71]. Alternatively, the data would also be compatible with a model in which strands are formed by unconventional lipid structures that are stabilised by junctional transmembrane proteins [71,72,73].

## 5. Conclusions

Our data thus provide new insights into the functions of ZO-1 in TJ assembly and epithelial morphogenesis, identifying tension-dependent and -independent roles. Our study indicates that reciprocal regulation between the mechanical properties of the cells and the forming TJ guides junction assembly and epithelial morphogenesis. ZO-1 is at the centre of this crosstalk by regulating signalling components such as GEF-H1. It will be important to determine how such functional properties of ZO-1 regulate dynamic biological processes and guide other junctional functions such as the regulation of gene expression.

## Data Availability

The datasets generated during and analysed during the current study are available from the corresponding authors on reasonable request.

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
