# Peer review of "ZO-1 Guides Tight Junction Assembly and Epithelial Morphogenesis via Cytoskeletal Tension-Dependent and -Independent Functions"

_cells, 2022, doi:10.3390/cells11233775_

Round 1
Reviewer 1 Report
This manuscript aims at providing new insights into the role of ZO-1 in controlling TJ assembly and epithelial morphogenesis, and explores a crosstalk between ZO-1, ECM stiffness and tension. Using ZO-1 depletion in MDCK cells, they show that the physical properties of the ECM determine how ZO-1 impacts AJ tension, actin cytoskeleton and TJ assembly. ZO-1 also affects cell morphology, including cell size, and focal adhesions and traction forces, and these effects are also impacted by ECM stiffness. ZO-1 depletion also disrupts normal TJ development and both bicellular and tricellular TJ ultrastructure. Further, ZO-1 and ZO-2 appear to have distinct roles on TJ regulation.
This is a strong study addressing an important question, namely the role of ZO-1, a key TJ adapter molecule. The manuscript provides new insight into the role of ZO-1 and provides information to resolves some contradictions in the literature. Specifically, the role of ZO-1 in various aspects of TJ assembly and function, and on cell morphology and mechanotransduction has been previously explored. However, studies from different cell systems provided contradictory results. This study offers an explanation for some of these discrepancies by providing proof that the effects of ZO-1 depends on ECM stiffness, and possibly the actin architecture of the cells. This conclusion will be useful for researchers working in the field, and ultimately contributes to our understanding of TJ regulation. They also compare the role of ZO-1 and that of ZO-2 to obtain new insights into the unique roles of these molecules, which again is an important aspect of understanding TJ regulation.
The study is detailed, with a large amount of well-presented data. Individual expreiments are well designed and controlled. The authors use sophisticated cellular systems to manipulate ZO-1 and ZO-2 multiple ways. They combined methods of modulating ECM stiffness with measuring various aspects of TJ development and functions. The measurements charactering the effects of ZO-1 depletion are well designed and thoroughly described. The images are extensively quantified to provide statistical analysis and strengthen conclusions. Overall, the study is technically sound and the data are conclusive.
There are however some weaknesses in the overall design and logic of the study. The study contains a large and diverse array of observations, and the logic is not always straightforward. In fact, at times the study becomes very complex, with too many variables (stiffness, cell density and ZO-1 depletion, phospho-myosin inhibition) that weakens the conclusiveness. The observations are not always connected or followed up, and this leads at times to overinterpretation. Especially the second half of the manuscript remains overly descriptive. More mechanistic insights would largely augment the conclusiveness of the study. Following are a couple suggestions and questions to help further enhance the final conclusions of the study.
1, Since ZO-1 appears to play a key role in determining the cell’s morphology on different substrates, it would be informative to show whether varying ECM stiffness affects ZO-1 expression/localization.
2, The authors conclude from the first figure (line 445-47) that “The correlation between F-actin organisation and impact of ZO-1 depletion on cell-cell tension supports a model in which the cytoskeletal architecture of different cell types determines the effect of ZO-1 depletion”. However, it is equally possible, that both (actin structure and the ultimate effects of ZO-1 depletion) are downstream from a common effect of ECM stiffness, and the actin structure does not determine how ZO-1 acts, but the two change in parallel. In fact, the relationship remains correlative, and there is no direct evidence that the effect of ZO-1 are indeed determined by the actin structure. Please refine this conclusion.
3, Fig 2. The findings are complicated by the fact that altering stiffness by itself also affects morphology: cells appear smaller on 40KPa and 1kPA than on glass surface. Please mention this. Also, please provide discussion on theories as to why there is no effect of ZO-1 depletion on morphology at low stiffness.
4, In Fig 3 the authors introduce yet another variable, namely cell density. What is the rationale for adding this variable? This part is hard to follow, quite complex, and less conclusive. What is the relationship between cell density and ECM stiffness, i.e. does cell density alter the effects of stiffness and through what mechanisms? Please provide a stronger rationale for introducing cell density as a variable, and a stronger explanation of the unique conclusions from these experiments, or consider moving the sparce/dense culture studies to supplemental figures to reduce the number of variables in this section. In Fig 3B: Is there a significant difference in blebbistatin-treated dense and sparce control samples (both on glass and 40kPa)? If so, what is the conclusion from this? In Fig 3C, D and E: what conditions were used to show ppMLC?
5, There is some disconnect between the first and second part of the paper. While the first part established the stiffness-dependent role of ZO-1 and shows that it controls tension and ppMLC, this aspect is not followed up in the later experiments. It would be good to address some aspects of the role of altered ppMLC in the downstream effects of ZO-1 depletion, and/or the mechanisms of increased ppMLC in ZO-1 depleted cells, as follows.
In Fig 5, do GEF-H1 and talin also affect ppMLC under the studied conditions? Does ppMLC mediate the effects of ZO-1 depletion on focal adhesion and traction? Fig 8 and 9: Does blebbistatin prevent the TJ disfunctions and ultrastructure changes observed following ZO-1 depletion? Adding these studies would strengthen the general conclusion that ZO-1 acts via ppMLC. Further, in the last couple of figures the effect of ECM stiffness has not been explored, even though this aspect is the key novel finding of the manuscript.
6, Fig 9C and D: On the figures please indicate using arrows the TJs, where present, to make it easier for the readers who are not familiar with these type of images to see the differences.
7, Fig S6A: the presence of claudin-4 in WT at the membrane is not well visible. Please provide separate images for claudin-4 and 2.
Author Response
Response to Reviewers
We would like to thank the reviewers for their supporting and detailed comments on our paper. We have made many changes to the manuscript to reply to their comments and added several new panels to clarify issues raised.
Both reviewers came to the impression that we concluded that all functions of ZO-1 in TJ assembly are related to cell tension. This is not the case as we had already summarised in the abstract and in various places throughout the manuscript (e.g., end of introduction, conclusions, etc.). Some of the requested experiments showing the effect of myosin inhibition or soft ECM on junction assembly in mature monolayers (i.e., discontinuous distribution of occludin) had also already been in the paper but apparently not in a sufficiently prominent manner and were missed. We have now made several changes to improve the clarity of the manuscript and added additional data in response to the reviewers’ comments. We also added a scheme in figure 12 that summarises the tension-dependent and -independent roles of ZO-1 in junction assembly.
Reviewer 1
This manuscript aims at providing new insights into the role of ZO-1 in controlling TJ assembly and epithelial morphogenesis, and explores a crosstalk between ZO-1, ECM stiffness and tension. Using ZO-1 depletion in MDCK cells, they show that the physical properties of the ECM determine how ZO-1 impacts AJ tension, actin cytoskeleton and TJ assembly. ZO-1 also affects cell morphology, including cell size, and focal adhesions and traction forces, and these effects are also impacted by ECM stiffness. ZO-1 depletion also disrupts normal TJ development and both bicellular and tricellular TJ ultrastructure. Further, ZO-1 and ZO-2 appear to have distinct roles on TJ regulation.
This is a strong study addressing an important question, namely the role of ZO-1, a key TJ adapter molecule. The manuscript provides new insight into the role of ZO-1 and provides information to resolves some contradictions in the literature. Specifically, the role of ZO-1 in various aspects of TJ assembly and function, and on cell morphology and mechanotransduction has been previously explored. However, studies from different cell systems provided contradictory results. This study offers an explanation for some of these discrepancies by providing proof that the effects of ZO-1 depends on ECM stiffness, and possibly the actin architecture of the cells. This conclusion will be useful for researchers working in the field, and ultimately contributes to our understanding of TJ regulation. They also compare the role of ZO-1 and that of ZO-2 to obtain new insights into the unique roles of these molecules, which again is an important aspect of understanding TJ regulation.
The study is detailed, with a large amount of well-presented data. Individual expreiments are well designed and controlled. The authors use sophisticated cellular systems to manipulate ZO-1 and ZO-2 multiple ways. They combined methods of modulating ECM stiffness with measuring various aspects of TJ development and functions. The measurements charactering the effects of ZO-1 depletion are well designed and thoroughly described. The images are extensively quantified to provide statistical analysis and strengthen conclusions. Overall, the study is technically sound and the data are conclusive.
There are however some weaknesses in the overall design and logic of the study. The study contains a large and diverse array of observations, and the logic is not always straightforward. In fact, at times the study becomes very complex, with too many variables (stiffness, cell density and ZO-1 depletion, phospho-myosin inhibition) that weakens the conclusiveness. The observations are not always connected or followed up, and this leads at times to overinterpretation. Especially the second half of the manuscript remains overly descriptive. More mechanistic insights would largely augment the conclusiveness of the study. Following are a couple suggestions and questions to help further enhance the final conclusions of the study.
We appreciate that the paper essentially contains two parts. The first part focuses on the stiffness-dependent role of ZO-1 and its interplay with cell mechanics, and was the more prominent part of the original manuscript as it contains more mechanistic information. However, the second part is required to present a complete characterisation of the ZO-1 knockout phenotype. Additionally, the phenotypes observed in high density cultures have previously not been described and challenge the current thinking of how ZO proteins function. We now added further experiments, moved some panels from the supplementary to the main figures, and better explained existing relevant panels that contain experiments with hydrogels and blebbistatin to document more clearly the differences between tension-dependent and independent effects (Fig. 6D, 7C, and 10C,D). We also added further experiments to clarify the reviewer’s questions.
1, Since ZO-1 appears to play a key role in determining the cell’s morphology on different substrates, it would be informative to show whether varying ECM stiffness affects ZO-1 expression/localization.
We have shown in a previous study that ECM stiffness regulates tension acting on ZO-1 but in cells plated on soft and stiff matrices, ZO-1 always localises to tight junctions. This is also supported by many published papers as ZO-1 on stiff substrates like plastic but also on very soft substrates (e.g., matrices for 3D morphogenesis assays, stiffness <1kPa) always localises to tight junctions (e.g., Terry et al., NCB 13, 159-166; paper shows ZO-1 stains in various cell types on glass and soft 3D gels; lines 448-450). We also added a panel in Figure S1D further illustrating this point and clarified the impact on ECM stiffness on ZO-1 in the introduction and in the discussion.
2, The authors conclude from the first figure (line 445-47) that “The correlation between F-actin organisation and impact of ZO-1 depletion on cell-cell tension supports a model in which the cytoskeletal architecture of different cell types determines the effect of ZO-1 depletion”. However, it is equally possible, that both (actin structure and the ultimate effects of ZO-1 depletion) are downstream from a common effect of ECM stiffness, and the actin structure does not determine how ZO-1 acts, but the two change in parallel. In fact, the relationship remains correlative, and there is no direct evidence that the effect of ZO-1 are indeed determined by the actin structure. Please refine this conclusion.
We agree with the reviewer that this is a correlation and not a direct proof. Hence, we previously finished the paragraph by stating that ‘The correlation between F-actin organisation and impact of ZO-1 depletion on cell-cell tension supports a model’. We have tried to further clarify this (lines 554 to 557). In the abstract, we have replaced ‘possibly via’ by ‘correlating with’ effects on the actin cytoskeleton (line 19). (Figure 1 is now Figure 4)
3, Fig 2. The findings are complicated by the fact that altering stiffness by itself also affects morphology: cells appear smaller on 40KPa and 1kPA than on glass surface. Please mention this. Also, please provide discussion on theories as to why there is no effect of ZO-1 depletion on morphology at low stiffness.
A corresponding sentence and reference has been added in the results section (lines 445-447) and the discussion has been extended (888 to 904).
4, In Fig 3 the authors introduce yet another variable, namely cell density. What is the rationale for adding this variable? This part is hard to follow, quite complex, and less conclusive. What is the relationship between cell density and ECM stiffness, i.e. does cell density alter the effects of stiffness and through what mechanisms? Please provide a stronger rationale for introducing cell density as a variable, and a stronger explanation of the unique conclusions from these experiments, or consider moving the sparce/dense culture studies to supplemental figures to reduce the number of variables in this section.
All experiments up to Figure 6 were done at the same cell density, plating the same number of cells for a sufficiently short time to exclude major effects of proliferation (see Materials and Methods, section 2.2 Experimental Setups, lines 102 to 122). As we stated on lines 436 to 441, these experiments aim to analyse the role of ZO-1 on epithelial morphogenesis during monolayer formation; hence, monolayers in a dynamic state had to be analysed. Only for figures 7 to 11, a different protocol was used as high density cells on filters were analysed to determine the effect on monolayers in equilibrium because of the discontinuous junctional distribution observed for occludin and claudins. Figure 2 (previous Figure 3) and other figures that show different densities show images from the centre of islands that are confluent and from the periphery that are less dense and, hence, more spread. Spread cells have a higher stress fibre content and, hence, higher junctional tension. The effect of myosin on rescue of junction formation is thus more striking at the periphery of cell islands as the cells have a stronger phenotype. There are no clear differences in cell density by the blebbistatin treatment as blebbistatin interferes with both junctional and basal actomyosin activity. The point here is that the cell shape is affected by the cytoskeletal tension cells can generate; hence, cells on stiff matrices spread more and, hence, appear less dense. ZO-1-depleted cells spread even more on a stiff matrix since ZO-1’s role in morphogenesis is to suppress cytoskeletal tension (hence, ZO-1 depletion induces increased traction along the basal membrane). We have now added further information about the known links between cell spreading and ECM stiffness in the introduction and tried to explain the conclusions better throughout the manuscript. We have also added figure numbers to the method section where we provide details about the experimental setups and design (lines 102 to 122) and tried to emphasis this important point throughout the manuscript.
In Fig 3B: Is there a significant difference in blebbistatin-treated dense and sparce control samples (both on glass and 40kPa)? If so, what is the conclusion from this? In Fig 3C, D and E: what conditions were used to show ppMLC?
As we explained above, as cells spread more on stiffer matrices due to increased actomyosin activity, blebbistatin will generally lead to less spreading.
5, There is some disconnect between the first and second part of the paper. While the first part established the stiffness-dependent role of ZO-1 and shows that it controls tension and ppMLC, this aspect is not followed up in the later experiments. It would be good to address some aspects of the role of altered ppMLC in the downstream effects of ZO-1 depletion, and/or the mechanisms of increased ppMLC in ZO-1 depleted cells, as follows.
In Fig 5, do GEF-H1 and talin also affect ppMLC under the studied conditions? Does ppMLC mediate the effects of ZO-1 depletion on focal adhesion and traction? Fig 8 and 9: Does blebbistatin prevent the TJ disfunctions and ultrastructure changes observed following ZO-1 depletion? Adding these studies would strengthen the general conclusion that ZO-1 acts via ppMLC. Further, in the last couple of figures the effect of ECM stiffness has not been explored, even though this aspect is the key novel finding of the manuscript.
The reviewer may have missed part of the supplementary data, which led to a misunderstanding. We did not claim that the phenotype observed in high density monolayers is actomyosin-dependent. In contrast, we had included Figure S5C showing that the discontinuous junctional distribution is not affected by the myosin inhibitor blebbistatin. Similarly, we had also shown that the discontinuous junctional phenotype is observed on soft and stiff ECM (Figure 6B).
To make the distinction between the tension-dependent and -independent function of ZO-1 clearer. We have now rearranged these data and moved the actomyosin inhibition experiment into the main figures (now Figure 7C). The old figure 7 had hence to be split into two to accommodate the additional data. We also tried to highlight the absence of effects of soft ECM and blebbistatin on the discontinuous occludin distribution in the description of these data (lines 655/658; 697-700). We also added additional data (Fig.10C,D), showing that the permeability changes of ZO-1 KO cells are not affected by blebbistatin (as expected form the absence of an effect of blebbistatin on TJ morphology in these high density cultures). We have now also added a summary scheme in figure 12, summarising the tension-dependent and tension-independent functions of ZO-1 in junction assembly. In relation to GEF-H1, we added panels in figure S5, showing a decrease in pMLC staining upon GEF-H1 depletion. There is no clear effect of talin depletion on myosin activation. However, talin functions as a cytoskeletal linker for integrins and is not directly involved in stimulation of myosin activity.
6, Fig 9C and D: On the figures please indicate using arrows the TJs, where present, to make it easier for the readers who are not familiar with these type of images to see the differences.
This is now figure 11. Arrows have been added.
7, Fig S6A: the presence of claudin-4 in WT at the membrane is not well visible. Please provide separate images for claudin-4 and 2.
Separate images have been provided.

Reviewer 2 Report
Overview:
In this manuscript the authors describe the effects of modifying stiffness of cell culturing systems on actin organization and have shown that ZO-1 is involved in regulating the cell’s response to ECM stiffness. The manuscript includes an impressive body of work and different techniques to examine the relationship between ZO-1 and ECM stiffness. The requirement for ZO-1 is studied in ZO-1-depleted cells as well as in two series of ZO-1/ZO-2 knockouts that were created in different strains of MDCK. The manuscript would benefit by some reorganization and streamlining of data and in some instances more details in the text and legends. Specific suggestions are outlined below.
Major comments:
1. The title of the manuscript is “Reciprocal regulation of cell mechanics and ZO-1 guides tight junction assembly and epithelial morphogenesis” suggests that there will be some balance in the number of experiments that look at both aspects of the reciprocal relationship. However, the majority of the experiments look at the effects of ZO-1 depletion in combination with the effects of culturing on different substrates on the presence of actin-cytoskeletal, tight junction and apical proteins. There are relatively few experiments that look at the effect of altered cell mechanics on ZO-1. The authors might consider altering the title to reflect the balance of experiments or including additional experiments that examine effects on ZO-1. For example, is there any effect on ZO-1 expression/localization in WT cells cultured in different ECM stiffnesses.
2. The experiments described in the manuscript use MDCK cells depleted for ZO-1 either using siRNA knockdown ZO-1 knockouts, to ensure complete absence of ZO-1. The manuscript would be more streamlined if parallel experiments were presented together to show the more severe phenotypes as the levels of ZO-1 are further decreased. One possibility is to begin the manuscript with figures that compare the different effects of ZO-1 knockdown and knockout on cell morphology, actin localization, and junctional protein staining. The functional experiments (TER, Dextran) could come immediately after, followed by follow-up experiments looking at cell tension (FRET, traction).
3. There appears to be some concern in the literature about using sensor modules incorporated in to E-cadherin, particularly for living cells (Eder, D., Basler, K. & Aegerter, C.M. Challenging FRET-based E-Cadherin force measurements in Drosophila . Sci Rep 7, 13692, 2017. https://doi.org/10.1038/s41598-017-14136-y). In the description of the results it is suggested that the tension at the basal surface is increasing, but it is not clear that the assay being used would measure this. Also, there is a lot of cytoplasmic signal, especially in the plastic and 40kPa images of Figure 1, panel D. Is E-cadherin affected by the culture systems?
4. As the cells grown on different substrates responded differently to ZO-1 knockdown, the rescue experiment shown in Figure 2 on glass should be repeated in the hydrogel conditions to see if there are different responses to the rescue and should be shown in a different panel.
5. The discussion of Figure 2 should include some information about cell density to relate to data from Figure 3. The cells in the 1kPa in 2D seem much smaller than cells in 1D relative to 20uM scale bar. Was there a difference in the total amount of sparse vs. dense areas between control and ZO-1 knockdown, suggesting an effect on tight junction formation? Is proliferative capacity of the cells changed on different surfaces? If yes, you might expect that this could affect cell size/density?
6. The images in Figure 4 A suggest that expression of mouse ZO-1 in MDCK cells transfected with siRNA control leads to smaller islands of cells than wild-type cells and this phenotype is attributed to altered proliferation. A proliferation assay should be performed to confirm this idea in the cells expressing mouse ZO-1 cells as well as the ZO-1 depleted cells in case the smaller island phenotype is also due to altered proliferation rates in the case of disrupted ZO-1. It is also not clear that this experiment is showing an effect on epithelial morphogenesis when the 3D cultures in supplemental would describe that better.
7. Line 501, p12 – effects on expression levels/localization of Occludin does not mean that the junctions are disrupted, or that the tight junctions are not formed (line 507, line 585). This should be made clearer, and it will be easier to follow if the functional experiments are discussed shortly before or immediately after. The fact that in some cases the effect on Occludin expression was greater in sparse areas is interesting.
8. The images in Figure 3A give the impression that there is less Occludin in the control siRNA of blebbistatin treated cells vs. the levels observed in siRNA CTL in vehicle-treated cells on all growth surfaces. Are these images representative and if so, the change in Occludin levels should be discussed. It also appears that in general the cells appear larger after blebbistatin treatment. This should be elaborated in the text. Either the ANOVA statistical test should be applied to the data of Figure B, G, H, or all t-test comparisons shown. The graphs should also be made larger for better readability.
9. The discussion of results from Figure 5 should be further expanded. Why are talin-depleted cells so much smaller? And depleting ZO-1 seems to rescue size, is this true?
Minor comments:
1. Methods – include information on z-section thickness
2. Line 488, p. 9 – issues with sentence structure, maybe replace “on” with “the”
3. Line 419-422, p9 – Are the results ‘contradictory’ or do they suggest that the effect of ZO-1 depletion is context dependent?
4. Figure 1, panel D and E – Labels need to be consistent. The image in D is labeled MG-Plastic, while the graphs show MG-Glass. There is no image in D corresponding to the measurements of Glass alone.
5. Figure 1, panel A – Needs more information in legend: What are blue and green fluorescence in right column? Describe what yellow and red dotted lines are indicating?
6. Supplemental Figure S1 – indicate approximate position of xz relative to xy image
7. Figure 2, panel D and E – ideally the images shown as examples of segmentation should be the same ones shown in D as there appears to be some inconsistencies. For example, the 1kPa siRNA ZO1 cells from panel D that are more irregular seem to be larger but this is not the impression from the 2D segmentation in E. Also add process used for segmentation in figure legend.
8. Figure 3 – Size of westerns could be increased in 3C and 3E also why aren’t data from 3E included in 3D?
9. Figure 3, panel F – Include arrows and zoomed insets to show examples of bundle vs. sharp lateral ppMLC staining.
10. Figure 3, panel I – Use z-sections to show the example of cell flattening to explain the weaker signal.
11. Line 514, p12 – make it clearer that there is no effect on localization of ppMLC but there is a change in IF signal and by western
12. Line 552, p. 14 – issues with sentence structure
13. Line 568, p14 – needs more justification. Previous results showed no change in AJs, so why would you follow up by looking at focal adhesions?
14. Figure 5 – make it clear that control and ZO-1 labels are referring to siRNA in panel D-F
15. Discuss the phenotype of DKO ZO-1/2 grown on 1kPA of Figure 6 in text
16. Why are both the ZO-1 KO C1 vs C2 first shown in Figure 6? Unless more information is added in the discussion about the difference between them, the other should be in supplemental.
17. The GFP-ZO-2 signal in Figure 7C is mostly cytoplasmic, and therefore cannot be considered a rescue experiment. Show the GFP-ZO-2 staining in control cells.
18. Figure 7 – include culture time in figure caption and describe what box areas are in panel A
19. Figure 7 – Cingulin stained cells have different morphology, is this due to different Z-stacks?
20. When possible include both apical and z-section images to better illustrate the immunofluorescence signal (e.g. Figure 8 panel A, B)
21. Figure 8 – the change in nuclear position should be quantified and discussed further. Has this been observed before?
22. The full TER data (all timepoints) and dextran staining from Figure 8 should be included in the supplemental figures
23. Supplemental Figure 5 – missing proper figure legend. What is being stained in panel C?
24. Figure 9, panels C & D – it would be helpful to have arrowheads or arrows indicating tight junction strands.
Author Response
We would like to thank the reviewers for their supporting and detailed comments on our paper. We have made many changes to the manuscript to reply to their comments and added several new panels to clarify issues raised.
Both reviewers came to the impression that we concluded that all functions of ZO-1 in TJ assembly are related to cell tension. This is not the case as we had already summarised in the abstract and in various places throughout the manuscript (e.g., end of introduction, conclusions, etc.). Some of the requested experiments showing the effect of myosin inhibition or soft ECM on junction assembly in mature monolayers (i.e., discontinuous distribution of occludin) had also already been in the paper but apparently not in a sufficiently prominent manner and were missed. We have now made several changes to improve the clarity of the manuscript and added additional data in response to the reviewers’ comments. We also added a scheme in figure 12 that summarises the tension-dependent and -independent roles of ZO-1 in junction assembly.
Reviewer 2
Overview:
In this manuscript the authors describe the effects of modifying stiffness of cell culturing systems on actin organization and have shown that ZO-1 is involved in regulating the cell’s response to ECM stiffness. The manuscript includes an impressive body of work and different techniques to examine the relationship between ZO-1 and ECM stiffness. The requirement for ZO-1 is studied in ZO-1-depleted cells as well as in two series of ZO-1/ZO-2 knockouts that were created in different strains of MDCK. The manuscript would benefit by some reorganization and streamlining of data and in some instances more details in the text and legends. Specific suggestions are outlined below.
Major comments:
- The title of the manuscript is “Reciprocal regulation of cell mechanics and ZO-1 guides tight junction assembly and epithelial morphogenesis” suggests that there will be some balance in the number of experiments that look at both aspects of the reciprocal relationship. However, the majority of the experiments look at the effects of ZO-1 depletion in combination with the effects of culturing on different substrates on the presence of actin-cytoskeletal, tight junction and apical proteins. There are relatively few experiments that look at the effect of altered cell mechanics on ZO-1. The authors might consider altering the title to reflect the balance of experiments or including additional experiments that examine effects on ZO-1. For example, is there any effect on ZO-1 expression/localization in WT cells cultured in different ECM stiffnesses.
The original title was chosen since the paper studies effects of ECM stiffness on the importance of ZO-1 in TJ formation. ECM stiffness directly regulates cytoskeletal tension and cell spreading, and we show that those effects that are ECM stiffness dependent can be rescued by myosin inhibition. However, the title obviously was misleading as the reviewers concluded that all functions of ZO-1 are tension-dependent. This is incorrect. Hence, we have now changed the title so that it is clear from the start that ZO-1 has tension-dependent and -independent functions.
- The experiments described in the manuscript use MDCK cells depleted for ZO-1 either using siRNA knockdown ZO-1 knockouts, to ensure complete absence of ZO-1. The manuscript would be more streamlined if parallel experiments were presented together to show the more severe phenotypes as the levels of ZO-1 are further decreased. One possibility is to begin the manuscript with figures that compare the different effects of ZO-1 knockdown and knockout on cell morphology, actin localization, and junctional protein staining. The functional experiments (TER, Dextran) could come immediately after, followed by follow-up experiments looking at cell tension (FRET, traction).
We have rearranged the manuscript to some extent and moved figures 1 to 3 to the start and follow them up with the tension measurements at cell-cell junctions and cell-ECM adhesion (figures 4 and 5). We did not follow the reviewer’s proposal as it would lead to a mix of different experimental strategies designed to study tension-dependent and -independent functions of ZO-1, which would be confusing. We also left figure 6 as it serves as a transition to the tension-independent functions in the second part that were studied in mature monolayers. The effects there (TER, dextran permeability, defects in junction assembly, etc.) are not tension-dependent. Hence, we left them separated to avoid confusion. We also added in the method section further clarifications to the section describing the experimental setup to clarify the protocols of the different experiments (Experimental setups: lines 112 to 134) .
- There appears to be some concern in the literature about using sensor modules incorporated in to E-cadherin, particularly for living cells (Eder, D., Basler, K. & Aegerter, C.M. Challenging FRET-based E-Cadherin force measurements in Drosophila. Sci Rep 7, 13692, 2017. https://doi.org/10.1038/s41598-017-14136-y). In the description of the results it is suggested that the tension at the basal surface is increasing, but it is not clear that the assay being used would measure this. Also, there is a lot of cytoplasmic signal, especially in the plastic and 40kPa images of Figure 1, panel D. Is E-cadherin affected by the culture systems?
The study by Eder et al. raises interesting observations about the use of a similar sensor in flies. The paper has a detailed discussions about the limitations of such sensors primarily in terms of dynamic range, which is a potential problem if no differences are observed (we do observe significant differences). They also speculate that E-cadherin may not be the relevant cell-cell adhesion protein in their system, which, being not Drosophila biologists, is not within our expertise to comment on.
It is difficult to compare the sensor used in the suggested paper with the one used in our manuscript as they have different designs (the sensor we used has the FRET module incorporated between the p120catenin and the b-catenin binding sites). Functionality of this sensor and its use in MDCK cells including responses to externally applied mechanical stress have been characterised and analysed in great detail by Borghi et al. (2012).
There is some cytoplasmic staining in all conditions. However, only FRET signals at the lateral membrane are analysed for the quantification.
- As the cells grown on different substrates responded differently to ZO-1 knockdown, the rescue experiment shown in Figure 2 on glass should be repeated in the hydrogel conditions to see if there are different responses to the rescue and should be shown in a different panel.
The only strong phenotype described in that figure (now figure 1, previously figure 2) is that there is decreased cell density in knockdown cells on glass, which became weaker on soft matrix. Hence, we rescued the strong phenotype. Additionally, subsequent experiments show that the KO cells induce the same effect on cell morphology. The effect on 1kPa is more evident in the monolayer morphogenesis experiments as the cells contract (now Fig.3) that show that the contraction on 1kPa leads to monolayer fragmentation. The figure also shows that this phenotype is not only rescued by myosin inhibition but also expression of RNAi-resistant GFP-mZO-1. Hence, we believe that our experiments appropriately test the specificity of the phenotypes induced by ZO-1 deficiency.
- The discussion of Figure 2 should include some information about cell density to relate to data from Figure 3. The cells in the 1kPa in 2D seem much smaller than cells in 1D relative to 20uM scale bar. Was there a difference in the total amount of sparse vs. dense areas between control and ZO-1 knockdown, suggesting an effect on tight junction formation? Is proliferative capacity of the cells changed on different surfaces? If yes, you might expect that this could affect cell size/density?
We first need to apologise for an error in the scale bar identification in figure 1D (now Fig.5D), which is 10μm. The substrates of these cells is in principle the same but there are of course some batch-to-batch variations. The experimental protocols in these experiments are different as the tension sensors need to be transiently transfected the day before the experiment; what may lead to some unavoidable differences in cell morphology. However, the important part here is that within a type of experiment, the cells were always treated the same way and all experiments were repeated with different batches of hydrogels.
- The images in Figure 4 A suggest that expression of mouse ZO-1 in MDCK cells transfected with siRNA control leads to smaller islands of cells than wild-type cells and this phenotype is attributed to altered proliferation. A proliferation assay should be performed to confirm this idea in the cells expressing mouse ZO-1 cells as well as the ZO-1 depleted cells in case the smaller island phenotype is also due to altered proliferation rates in the case of disrupted ZO-1. It is also not clear that this experiment is showing an effect on epithelial morphogenesis when the 3D cultures in supplemental would describe that better.
The figure is now 3A. We have already published that ectopic expression of ZO-1 inhibits proliferation MDCK cells (Balda et al. 2003; Ref 57). We have not suggested or concluded that ZO-1 depletion leads to reduced proliferation and smaller islands. ZO-1 depletion of 1kPa ECM leads to smaller islands since the cells contract. Hence, monolayer formation is rescued when cells are incubated with blebbistatin. The KO cells in figure 3D also make the same type of small islands, further supporting the specific of the phenotype. The experiments from Fig.1 to 6 make use of a short plating period (2 days) to avoid large differences in cell numbers interfering with the results (see Experimental Setup in Materials and Methods). We are currently performing proliferation assays to analyse the roles of ZO-1 in transcription using these KO clones and have not found any significant effect after such short time periods. Moreover, after longer periods culture (> 3days), KO cells proliferate faster not more slowly.
We did confirm that ZO-1 depletion also disrupts morphogenesis in 3D (now Fig.S3). However, this has already previously been established by different laboratories (Sourisseau et al., 2006; Odenwald et al, 2016). The point of the 2D morphogenesis assay is not to confirm the role of ZO-1 in morphogenesis but to determine if the morphogenesis defect can be caused by soft ECM.
- Line 501, p12 – effects on expression levels/localization of Occludin does not mean that the junctions are disrupted, or that the tight junctions are not formed (line 507, line 585). This should be made clearer, and it will be easier to follow if the functional experiments are discussed shortly before or immediately after. The fact that in some cases the effect on Occludin expression was greater in sparse areas is interesting.
We could not identify the specific sections the reviewer was referring to as the stated lines did not contain conclusions about tight junctions or occludin. However, we tried to phrase conclusions more carefully throughout the manuscript. Also, there must be a clear distinction made between discontinuous occludin staining and absence of any sort of junctional staining.
- The images in Figure 3A give the impression that there is less Occludin in the control siRNA of blebbistatin treated cells vs. the levels observed in siRNA CTL in vehicle-treated cells on all growth surfaces. Are these images representative and if so, the change in Occludin levels should be discussed. It also appears that in general the cells appear larger after blebbistatin treatment. This should be elaborated in the text. Either the ANOVA statistical test should be applied to the data of Figure B, G, H, or all t-test comparisons shown. The graphs should also be made larger for better readability.
This is now figure 2. The data are not normally distributed; hence, a nonparametric Kruskall-Wallis test was applied instead of an ANOVA. The resulting p-value is and was indicated above the x-axis of the respective graphs. The pairwise comparisons are based on nonparametric Wilcoxon tests. Over all substrates, the pairwise comparisons do not indicate significant changes between vehicle and blebbistatin treated control cells. Similarly, there is some variability in cell sizes and the blebbistatin affects the appearance of cell junctions, which may cause the impression of altered cell shapes.
- The discussion of results from Figure 5 should be further expanded. Why are talin-depleted cells so much smaller? And depleting ZO-1 seems to rescue size, is this true?
Talin depletion inhibits focal adhesion function; hence, the cells spread less and look smaller (Elosegui-Artola et al. 2016). ZO-1 depletion has the opposite effect and induces cell spreading (Fig.1). However, the talin depletion phenotype is very strong and, hence, the double knockdown looks essentially the same as the single talin knockout (compare with single ZO-1 knockdown). Therefore, we focused the quantification on junction formation: the purpose of this experiment was to determine whether basal adhesion impacts on junction formation in response to ZO-1 depletion.
Minor comments:
- Methods – include information on z-section thickness
Thickness has been added.
- Line 488, p. 9 – issues with sentence structure, maybe replace “on” with “the”
The reviewer may refer to line 418. The sentence was changed.
- Line 419-422, p9 – Are the results ‘contradictory’ or do they suggest that the effect of ZO-1 depletion is context dependent?
We are not suggesting anything here. We follow up results from different experimental approaches that report increases or decreases in cell-cell tension upon ZO-1 depletion. Our results now indicate that the differences may indeed by context dependent.
- Figure 1, panel D and E – Labels need to be consistent. The image in D is labeled MG-Plastic, while the graphs show MG-Glass. There is no image in D corresponding to the measurements of Glass alone.
The figure has been corrected as suggested.
- Figure 1, panel A – Needs more information in legend: What are blue and green fluorescence in right column? Describe what yellow and red dotted lines are indicating?
This is now Figure 4D. The RGB images show an overlay of F-actin with the Hoechst stain (hence, we had marked the image column with F-actin in green and Hoechst in blue). The dotted lines indicate the apical and basal monolayer edges.
- Supplemental Figure S1 – indicate approximate position of xz relative to xy image
The images were acquired as xz scans that were acquired randomly across the sample. They are not reconstitutions of serial xy sections, and, hence, the positions in xy cannot be reconstituted to a reasonable precision.
- Figure 2, panel D and E – ideally the images shown as examples of segmentation should be the same ones shown in D as there appears to be some inconsistencies. For example, the 1kPa siRNA ZO1 cells from panel D that are more irregular seem to be larger but this is not the impression from the 2D segmentation in E. Also add process used for segmentation in figure legend.
The panels shows a set of representative images of F-actin staining and samples of segmentations performed with junctional markers as an illustration of the quantification. They are representative for the data they represent. We have now also added an example image of F-actin segmentation in figure S1A.
- Figure 3 – Size of westerns could be increased in 3C and 3E also why aren’t data from 3E included in 3D?
The blot sizes have been increased as suggested. Hydrogel samples are difficult to obtain in sufficiently large amounts to obtain quantitatively reliable blots of regulatory components such as MLC. While they do show increases in agreement with the immunofluorescence, we did not think they are appropriate to calculate quantitative changes.
- Figure 3, panel F – Include arrows and zoomed insets to show examples of bundle vs. sharp lateral ppMLC staining.
The figure has been enlarged as far as possible as insets would cover too much of the images. Arrows and arrowheads have been added to label sharp lateral actomyosin and peripheral bundles.
- Figure 3, panel I – Use z-sections to show the example of cell flattening to explain the weaker signal.
All what we concluded from this panel (now Fig.2I) is that there was no consistent effect on p120catenin staining. We made no conclusions about weaker and stronger signals. These are epifluorescence images and no meaningful z-sections can be generated from those.
- Line 514, p12 – make it clearer that there is no effect on localization of ppMLC but there is a change in IF signal and by western
Sentence has been clarified.
- Line 552, p. 14 – issues with sentence structure
Corrected.
- Line 568, p14 – needs more justification. Previous results showed no change in AJs, so why would you follow up by looking at focal adhesions?
An additional sentence was added.
- Figure 5 – make it clear that control and ZO-1 labels are referring to siRNA in panel D-F
Corrected.
- Discuss the phenotype of DKO ZO-1/2 grown on 1kPA of Figure 6 in text
Has been added.
- Why are both the ZO-1 KO C1 vs C2 first shown in Figure 6? Unless more information is added in the discussion about the difference between them, the other should be in supplemental.
There is no difference between the two clones. We think it is important to show that different knockout clones exhibit the same phenotype, particularly as it is different to what has previously been observed in KOs not resulting in efficient depletion.
- The GFP-ZO-2 signal in Figure 7C is mostly cytoplasmic, and therefore cannot be considered a rescue experiment. Show the GFP-ZO-2 staining in control cells.
We did not conclude that GFP-ZO-2 rescues TJ formation and is efficiently recruited to junctions. It behaves like endogenous ZO-2 in ZO-1 KOs (see Fig.7B), which is what one would expect. An image of expression of GFP-ZO-2 in wild-type cells has been added in Fig.S7C, and shows that the construct is indeed recruited to TJ in wild-type cells.
- Figure 7 – include culture time in figure caption and describe what box areas are in panel A
The details have been added to the legend.
- Figure 7 – Cingulin stained cells have different morphology, is this due to different Z-stacks?
It reflects variability in cell morphology within the monolayers.
- When possible include both apical and z-section images to better illustrate the immunofluorescence signal (e.g. Figure 8 panel A, B)
We cannot fit all these images in the main figures. So, the xy images of claudin-2 for which we show xz-scans in the main figure are in Fig.S8. Additionally, some of the claudin antibodies don’t stain brightly enough to generate meaningful z sections particularly in knockout cells where the staining is not concentrated anymore at cell-cell junctions.
- Figure 8 – the change in nuclear position should be quantified and discussed further. Has this been observed before?
The quantification is now shown in panel 9E and is referred to in the text. No, this was not observed previously.
- The full TER data (all timepoints) and dextran staining from Figure 8 should be included in the supplemental figures
The TER kinetic is now shown in Fig.S9. Fluorescent dextran diffusion is measured by adding labelled dextrans to the medium on one side of monolayers and then measuring how much crosses to the medium on the opposite side. The measurements are done with a fluorometer; hence, there are no ‘stainings’ that could be shown.
- Supplemental Figure 5 – missing proper figure legend. What is being stained in panel C?
Figure legend has been completed.
- Figure 9, panels C & D – it would be helpful to have arrowheads or arrows indicating tight junction strands.
Arrows for bicellular junction strands and arrowheads for strands in tricellular junctions have been added.
Round 2
Reviewer 1 Report
I would like to thank the authors for addressing my concerns. I feel that the revisions have improved the manuscript. While I feel that the study remains rather complex, I think that the changes have improved it, Importantly, the manuscript contains a large array of interesting data and the study in general is well designed and conclusive, and will be of interest to the field. I have no further specific recommendations.